# Dissecting esophageal squamous-cell carcinoma ecosystem by single-cell transcriptomic analysis

Xiannian Zhang [1,11], Linna Peng[2,11], Yingying Luo [2,11], Shaosen Zhang[2,11], Yang Pu[3,11], Yamei Chen [2,11], Wenjia Guo [4,5], Jiacheng Yao [6], Mingming Shao[2], Wenyi Fan [2], Qionghua Cui [2], Yiyi Xi[2], Yanxia Sun [2], Xiangjie Niu[2], Xuan Zhao [2], Liping Chen[2], Yuqian Wang [2], Yachen Liu[2], Xinyu Yang[2], Chengcheng Wang[2], Ce Zhong[2], Wen Tan[2], Jianbin Wang [6✉], Chen Wu [2,7,8,9✉] & Dongxin Lin [2,7,9,10]

Esophageal squamous-cell carcinoma (ESCC), one of the most prevalent and lethal malignant disease, has a complex but unknown tumor ecosystem. Here, we investigate the composition of ESCC tumors based on 208,659 single-cell transcriptomes derived from 60 individuals. We identify 8 common expression programs from malignant epithelial cells and discover 42 cell types, including 26 immune cell and 16 nonimmune stromal cell subtypes in the tumor microenvironment (TME), and analyse the interactions between cancer cells and other cells and the interactions among different cell types in the TME. Moreover, we link the cancer cell transcriptomes to the somatic mutations and identify several markers significantly associated with patients' survival, which may be relevant to precision care of ESCC patients. These results reveal the immunosuppressive status in the ESCC TME and further our understanding of ESCC.

[1] School of Basic Medical Sciences, Beijing Advanced Innovation Center for Human Brain Protection, Capital Medical University, Beijing, China. [2] Department of Etiology and Carcinogenesis, National Cancer Center/Cancer Hospital, Chinese Academy of Medical Sciences and Peking Union Medical College, Beijing, China. [3] Institute of Basic Medical Sciences, Chinese Academy of Medical Sciences and Peking Union Medical College, Beijing, China. [4] Cancer Institute, Affiliated Cancer Hospital of Xinjiang Medical University, Urumqi, China. [5] Key Laboratory of Oncology of Xinjiang Uyghur Autonomous Regio, Urumqi, China. [6] School of Life Sciences, Tsinghua-Peking Center for Life Sciences, Tsinghua University, Beijing, China. [7] Collaborative Innovation Center for Cancer Personalized Medicine, Nanjing Medical University, Nanjing, China. [8] CAMS Oxford Institute (COI), Chinese Academy of Medical Sciences, Beijing, China. [9] CAMS key Laboratory of Cancer Genomic Biology, Chinese Academy of Medical Sciences and Peking Union Medical College, Beijing, China. [10] Sun Yat-sen University Cancer Center, State Key Laboratory of Oncology in South China, Guangzhou, China. [11] These authors contributed equally: Xiannian Zhang, Linna Peng, Yingying Luo, Shaosen Zhang, Yang Pu, Yamei Chen. ✉email: jianbinwang@tsinghua.edu.cn; chenwu@cicams.ac.cn

Esophageal cancer ranks the sixth leading cause of cancer-related death worldwide[1] with esophageal squamous-cell carcinoma (ESCC) accounting for over 90% of total esophageal cancer in China. ESCC has poor prognosis with the 5-year survival rates around 20–30% probably due to the difficulty in early diagnosis and the lack of effective therapy[1–3]. Deciphering the complexity of ESCC tumor and its microenvironment is therefore fundamental for establishing early diagnosis and creating effective and precision treatment. In recent years, whole-genome/exome sequencing analysis on bulk tumor tissues has suggested that mutations of certain genes, such as *TP53* and *NOTCH1*, or aberrant pathways, such as cell cycle and PI3K-AKT, may drive the development of ESCC[4–7]. However, it seems that genomic alterations are not completely attributed to the initiation and progression of ESCC and failed to translate into clinical use because such alterations are also seen in aging but normal epithelium[8,9].

In addition, the tumor microenvironment (TME) compositions including various immune cells and nonimmune stromal cells also play determinant roles in tumor growth or decline[10–12]. It has been shown that immune cells and other stromal cells in TME rarely have genomic alterations[13–16]; however, they might interact with cancer cells to affect tumor progression and anticancer treatment[17–19]. For example, the checkpoint blockade immunotherapy can only benefit limited part of cancer patients, due to the cancer-immune interplays[20–22]. The failure of several anti-angiogenic therapies for cancer has been believed to be due to the metabolic adaptation and reprogramming of cancer cells and the abnormality of endothelial cells and their interaction with pericytes[23].

Transcriptomic aberrancies are also critical events in promoting cancer development and progression[24–26]. In the last decades, many transcriptomic studies on ESCC have been reported, but previous studies were relied on bulk-tissue-based cDNA microarrays or RNA-sequencing analysis[4,5,27–29]. The bulk-tissue-based transcriptome analysis detects a mixed transcriptome derived from both cancer cells and TME components and usually generates massive marker genes in diversified pathways, which do not tell the dynamic status of various cells and their interactions during the development of cancer in the heterogeneous tumor tissues. In this regard, transcriptome analysis using bulk tumor tissues seems not suitable for the purpose of deciphering various unknown molecular events at the transcriptome level from different cells involving in the progression of ESCC.

High-throughput single-cell RNA sequencing (scRNA-seq), a valid method developed in recent years, has been proved to enable the dissection of heterogeneous tumors and deciphering the interaction between cancer cells and their microenvironment components[30–34]. Elucidating the transcriptome characteristics of cancer cells and the microenvironment components and their interactions, which are largely unknown in ESCC, is basic and fundamental in further understanding the cancer and developing effective early diagnosis and treatment strategies.

In the present study, we have performed scRNA-seq on a large scale of cells derived from ESCC tumors obtained from 60 patients to decode the transcriptome alterations associated with the cancer. We have also performed whole-exome/genome sequencing on bulk tissues from the same ESCC tumors and incorporated genomic data for analysis. We have dissected 8 expression programs from malignant epithelial cells and decomposed the TME compositions into 42 functional subtypes. By integrating these results, we have established a primary association framework between cancer cells and various noncancerous cells in the TME, which contributes to ESCC progression and prognosis.

## Results

**Overview of ESCC ecosystem characterized by scRNA-seq.** To decipher the cell composition within the ESCC tumors, we performed scRNA-seq and T cell receptor (TCR)-seq (for CD45+ cells only) on 60 ESCC tumor and 4 adjacent normal tissue samples obtained from 60 individuals using 10X Genomics platform (Fig. 1a; Supplementary Fig. 1a; Supplementary Data 1a and 2). After quality controls, we retained single-cell transcriptome of 208,659 cells including 97,631 CD45− and 111,028 CD45+ cells. Following regressing against read depth and mitochondrial read count, we performed graph-based clustering of the combined CD45− and CD45+ dataset and annotated the clusters using established marker genes (Fig. 1b, c; Supplementary Fig. 1b, c). We identified 8 main cell populations: epithelial cells ($N = 44,730$), fibroblasts ($N = 37,213$), endothelial cells ($N = 11,267$), pericytes ($N = 3102$), and fibroblastic reticular cells (FRC; $N = 1,319$) from CD45− dataset and T cells ($N = 69,278$), B cells ($N = 22,477$) and myeloid cells ($N = 19,273$) from CD45+ dataset. The epithelial cells exhibited high expression level of classic epithelial markers including *EPCAM*, *SFN*, and cytokeratins, and high genomic instability as demonstrated by severe copy number variations (CNVs) inferred from the transcriptome dataset and bulk whole-genome sequencing (WGS) results (Supplementary Fig. 1d–g), indicating that most epithelial cells are malignant. We then systematically analyzed the ecosystem compositions of each ESCC and found that the cell-type proportions were highly variable within and across patients, with some variations being associated with tissue type and tumor stage (Supplementary Fig. 1h). ESCC tumors had more epithelial cells and pericytes but fewer fibroblasts than adjacent normal tissues ($P < 0.05$; Fig. 1b; Supplementary Fig. 1i; Supplementary Data 3 and 4). The proportion of B cells was significantly less in stage II/III ESCC tumors compared with that in stage I tumors ($P = 0.013$; Supplementary Fig. 1j). These results suggest that ESCC ecosystems are highly heterogeneous, which should be further deciphered in the following analysis.

**Correlation between intra- and inter-tumor heterogeneity of epithelial cells.** We then investigated whether and how the expression states varied among epithelial cells. By analyzing the transcriptome patterns of epithelial cells in 52 tumors that had >100 epithelial cells, we found that they could be divided into 38 clusters (Fig. 2a). We then analyzed the patient contributions to each cluster and found that 24 clusters had cells (≥75%) from sole individual patients while other 14 clusters had the expression patterns shared by multiple patients (Fig. 2b). We defined the 24 clusters as Group 1 clusters (with ≥75% personal cells) and the remaining as Group 2 clusters. We found Group 1 clusters showed increased pathway activities of cell proliferation and EMT, while Group 2 clusters presented activation of immunity-related pathways such as the complement, inflammatory, and IL2/STAT5 signaling (Supplementary Fig. 2a). We also compared the epithelial composition in each ESCC and found that ESCC from 21 patients were mostly composed of Group 1 clusters (≥60%) and ESCC from 31 patients were comprised of multiple Group 2 clusters (Fig. 2c). These results indicate a pervasive intra- and inter-tumor heterogeneity in ESCC.

We used principal components to measure the intra- and inter-tumor heterogeneity levels of epithelial cells. The intra-tumor heterogeneity is defined by the dispersion of cells from average within each sample, while inter-tumor heterogeneity is characterized by the distance of cells between each sample and global average as illustrated in Fig. 2d (see "Methods"). We integratedly quantified the levels of intra- and inter-tumor heterogeneity

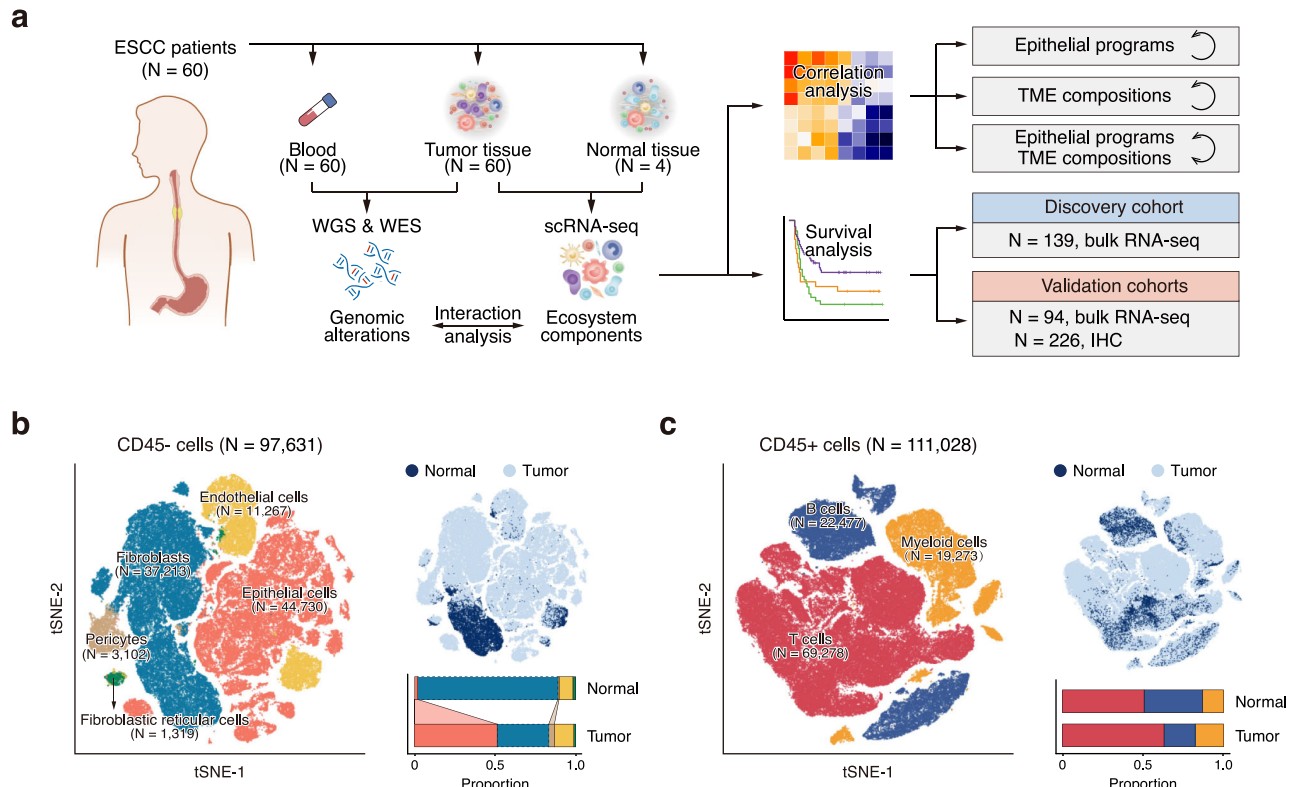

**Fig. 1 Overview of ESCC ecosystem characterized by scRNA-seq. a** Scheme of the overall study design. **b**, **c** tSNE plots of 97,631 CD45- cells (**b**) and 111,028 CD45+ cells (**c**), colored by cell type (*left*) and tissue type (*top right*). The proportions of cell types in normal and tumor tissues are shown on the bottom right (colors as in the left panel). Cell types that significantly increase (solid line filled with color) or decrease (dashed line) in tumor tissue (two-sided Wilcoxon test, $P < 0.05$) are indicated by links between normal and tumor bars. $P$ values: 0.001 (epithelial cell), 0.001 (fibroblast), 0.010 (pericyte).

and found that they were positively correlated ($r = 0.64$, $P = 7.3 \times 10^{-7}$; Fig. 2e; Supplementary Fig. 2b, c). Among patients with the same intra-tumor heterogeneity levels, those having the Group 1 cluster had relatively higher inter-tumor heterogeneity levels than those without Group 1 cluster ($P = 1.2 \times 10^{-7}$; Fig. 2e, f; Supplementary Fig. 2d). The relative heterogeneity level was associated with age and drinking status of ESCC patients (Supplementary Fig. 2e). The relative heterogeneity level in ESCC with age >60 was significantly higher than that in ESCC with age ≤60 ($P = 0.041$), and it was also higher in ESCC from non-drinkers than that in ESCC from drinkers ($P = 0.048$).

**Identification of eight common expression programs of epithelial cells in ESCC tumors.** We developed a meta-cluster approach to uncover co-expressed programs among malignant cells from multiple samples. We first clustered the epithelial cells within each sample and generated a total of 274 intra-tumor subclusters in 52 ESCC tumors. We then defined an expression module for each subcluster with preferentially expressed 30 genes and subsequently aggregated the 274 modules into multiple recurrent expression programs using hierarchical clustering based on their expression profiles. We identified 8 expression programs with different functions and cell status (Fig. 3a, b; Supplementary Fig. 2f, g; Supplementary Data 5) and defined cells expressing ≥70% of genes in a given program as program cells (Supplementary Fig. 2h). We selected the most activated pathways in the program cells by comparison with non-program cells to analyze their functions (Fig. 3c). The mucosal immunity-like (Mucosal) program was characterized by the expression of genes associated with innate immune response (e.g., *S100P*) and mucosal defensive mechanisms including mucosal chemokine (e.g., *CXCL17*) and mucus production (e.g., *AGR2* and *MUC20*). Mucosal cells

showed activation of the complement, TNFα signaling, apoptosis, and inflammatory response pathways. The stress responses (Stress) program consisted of immediate early genes (e.g., *EGR1*, *JUN*, and *FOS*) that are activated in response to widespread cellular stimuli and displayed upregulation of TNFα signaling, UV response, p53, and apoptosis pathways. The antigen presentation (AP) program had increased expression of major histocompatibility complex (MHC) class II molecules (e.g., *CD74*, *HLA-DPA1*, and *HLA-DRA/B1/B5*) that are involved in initiating adaptive antitumor immune responses. Immunity-related pathways, such as the allograft rejection, IFN-γ, IFN-α response, and the complement pathway, were activated in AP cells, probably indicating the reactivity to tumor neoantigens. The cell cycle (Cycling) program was characterized by high expression of genes involved in cell proliferation (e.g., *CENPW*, *CKS1B*, and *BIRC5*) and presented activation of the E2F targets, G2M checkpoint and MYC targets pathways, suggesting tumor cell proliferation. Two programs were associated with epithelial differentiation (Epi1/2). The Epi1 program was characterized by the expression of stress keratins (*KRT6*, *KRT16*, and *KRT17*) that are associated with keratinocyte hyperproliferation and therefore may play a role in enhancing tumorigenesis and tumor growth[35–37]. Epi1 cells displayed upregulation of pathways involved in cell growth and proliferation, such as mTORC1 signaling and PI3K/AKT/mTOR signaling pathways, as well as metabolic pathways including oxidative phosphorylation and glycolysis. The Epi2 program had the overexpressed genes related to the terminal differentiation such as envelope proteins (*SPRR1A/1B*) and calprotectin (*S100A8/9*), apical surface, the PI3K/AKT/mTOR signaling, the complement, and p53 pathways. The mesenchymal cell-like (Mes) program consisted of genes such as *VIM* and *SPARC* and showed activation of epithelial-mesenchymal

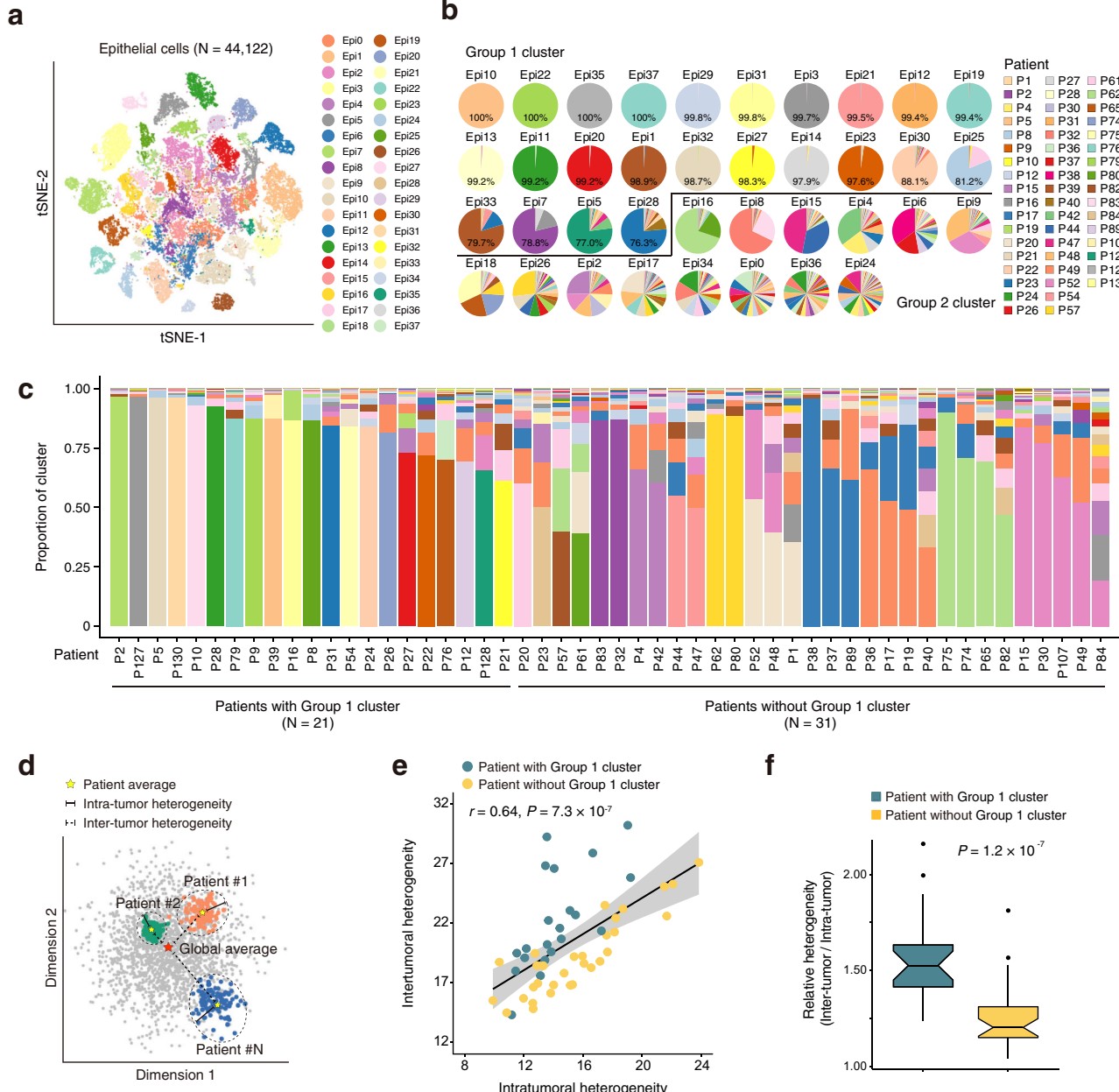

**Fig. 2 Correlation between intra- and inter-tumor heterogeneity of epithelial cells. a** tSNE plot of 44,122 epithelial cells, colored by cluster. Only tumors with at least 100 epithelial cells are shown. **b** Pie charts showing the proportions of cells originating from each patient detected in each cluster, colored by patient. Clusters are ordered by the proportion of the most dominant patient and are classified into Group 1 cluster or Group 2 cluster. **c** Stacked histograms showing proportions of epithelial cell clusters for each patient, colored by cluster as in (**a**). **d** Schematic showing the methods for quantifying the intra-tumor (solid line) and inter-tumor (dashed line) heterogeneities of epithelial cells for each patient based on the patient average (yellow star) and global average (red star). **e** Scatterplot showing the positive correlation between the intra- and inter-tumor heterogeneities. Two-sided Spearman correlation coefficient and P value are indicated. Shaded region indicates 95% confidence interval for the correlation. **f** Boxplot showing the relative inter-tumor heterogeneity levels for patients with Group 1 cluster (green) or without Group 1 cluster (yellow). Boxplots show the median (central line), the 25–75% interquartile range (IQR) (box limits), the ±1.5 times IQR (Tukey whiskers).

transition (EMT) and angiogenesis pathways. Finally, the oxidative stress or detoxification (Oxd) program was characterized by the expression of multiple peroxidases and reductases (e.g., *GPX2* and *AKR1C1*) involved in the defense against oxidative damage.

We performed the dependency analysis to infer the pairwise interactions among the expression programs and identified 5 significant co-occurring program pairs as well as 7 mutually exclusive program pairs with odds ratios ≥2 and ≤0.5, respectively (all $P < 0.05$) (Fig. 3d). The Epi1 was highly co-occurred with the Epi2, probably reflecting the need of both Epi1 and Epi2 for

epithelial cell differentiation. The Epi1 program was also co-occurred with the Cycling program, probably reflecting the activation of cell growth and proliferation pathways in Epi1 cells. The Epi2 program was highly co-occurred with the Mucosal program, suggesting that mucosal immunity is present in well-differentiated epithelial cells. The Mucosal program was exclusive of the Cycling and Mes programs, while the Mes program was exclusive of the Epi1, Epi2, and Oxd programs. The AP program was exclusive of the Epi2 and Oxd programs. We further measured the activity of these programs in each ESCC by

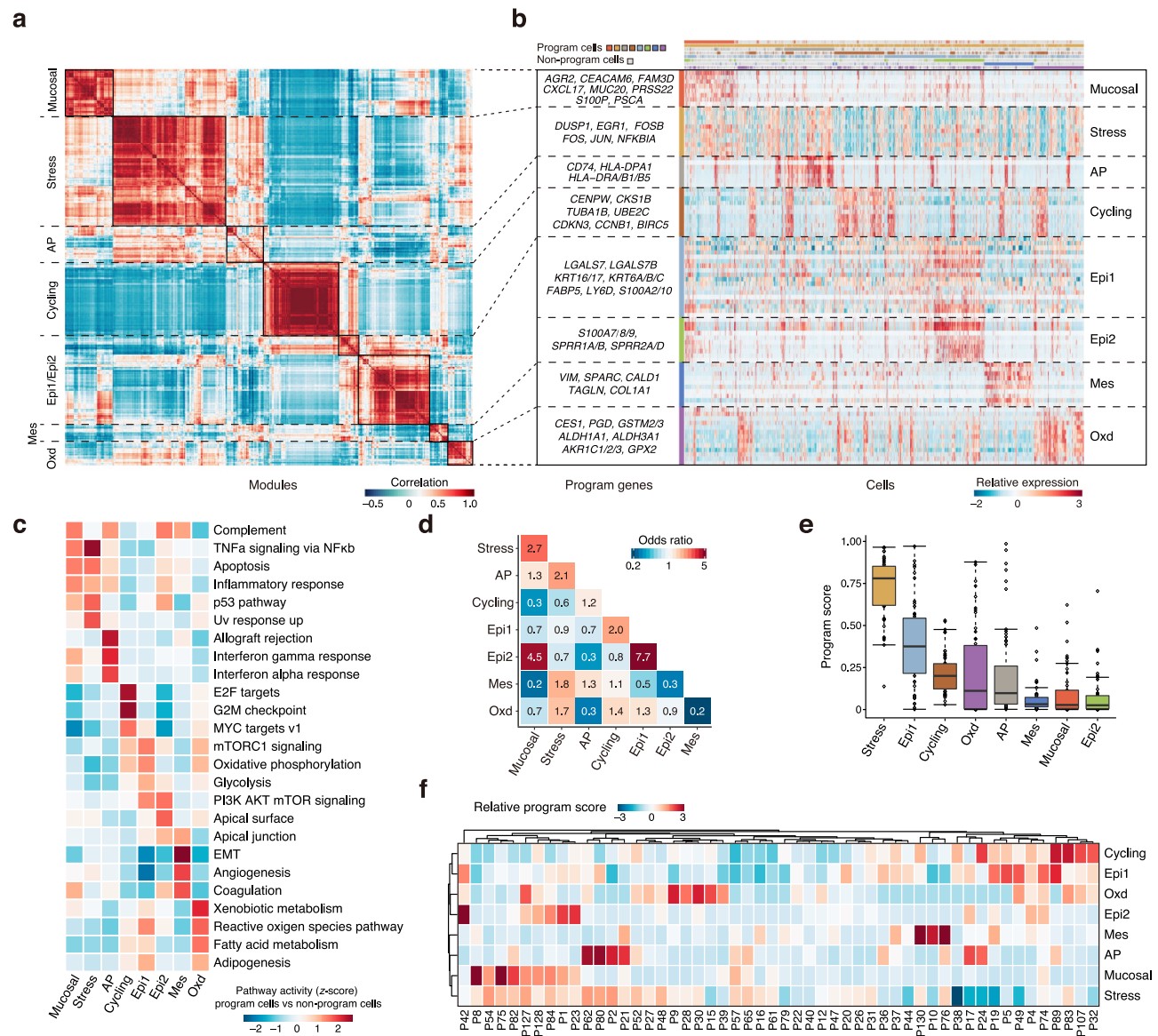

**Fig. 3 Identification of eight common expression programs of epithelial cells in ESCC tumors. a** Heatmap showing pairwise correlations of 274 modules derived from 52 tumors. The common expression programs across tumors are aggregated into clusters. **b** Heatmap showing expression of genes within each program across single cells. Randomly selected 500 cells for each program are shown. Colors above columns correspond to cell state (see Methods). **c** Heatmap showing differences in pathway activities between program cells and non-program cells for each program scored by GSVA. Each column is normalized by z-score to indicate the relative pathway activities. **d** Heatmap showing odds ratios assessing for each pair of programs (rows, columns) if they are co-occurrent ($\geq 2$, red) or exclusive ($\leq 0.5$, blue) than expected by chance ($P < 0.05$). $P$ values are derived from pairwise Fisher's exact test. **e** Boxplot showing the proportion of program cells (hereafter refers to as program score) for each tumor ($N = 52$) among 8 expression programs, sorted by the median program score. Boxplots show the median (central line), the 25–75% interquartile range (IQR) (box limits), and the ±1.5 times IQR (Tukey whiskers), and all data points, among which the lowest and the highest points indicate minimal and maximal values, respectively. **f** Clustered heatmap showing the normalized program scores for all programs in each tumor.

quantifying the program scores as proportion of corresponding program cells (Fig. 3e). Patients were then hierarchically clustered based on the program scores (Fig. 3f). We observed that most ESCC tumors contained the epithelial cells that expressed more than one program and that the Epi1 scores were highly variable among tumors with median score = 0.38 (ranging from 0 to 0.97). Patients with high Epi1 score usually had high Cycling score. The Epi2 and Mucosal programs had very small activities with the median scores being 0.03 among patients. Together, these results indicate that epithelial cells in ESCC tumors had 8 expression programs and the activity of each program was highly heterogeneous among tumors.

**Characterization of immunosuppressive ESCC tumor microenvironment.** Since T cells are the most abundant tumor-infiltrating lymphocytes (TILs) and highly heterogeneous in the TME, dissecting T cell populations would allow the accurate characterization of the immune status in tumors. We re-clustered 69,278 T cells and identified 9 distinct T cell subtypes and phenotypically related natural killer (NK)/natural killer T (NKT) cells (Fig. 4a; Supplementary Fig. 3a, b; Supplementary Data 6a). T cell subtypes included naïve T ($T_N$) cells, T helper 17 ($T_H$17) cells, follicular helper T ($T_{FH}$1/2) cells, regulatory T ($T_{reg}$) cells, memory T ($T_{MEM}$-CD4/CD8) cells, effector T ($T_{EFF}$) cells and exhausted T ($T_{EX}$) cells. Compared with $T_{FH}$2 cells, $T_{FH}$1 cells

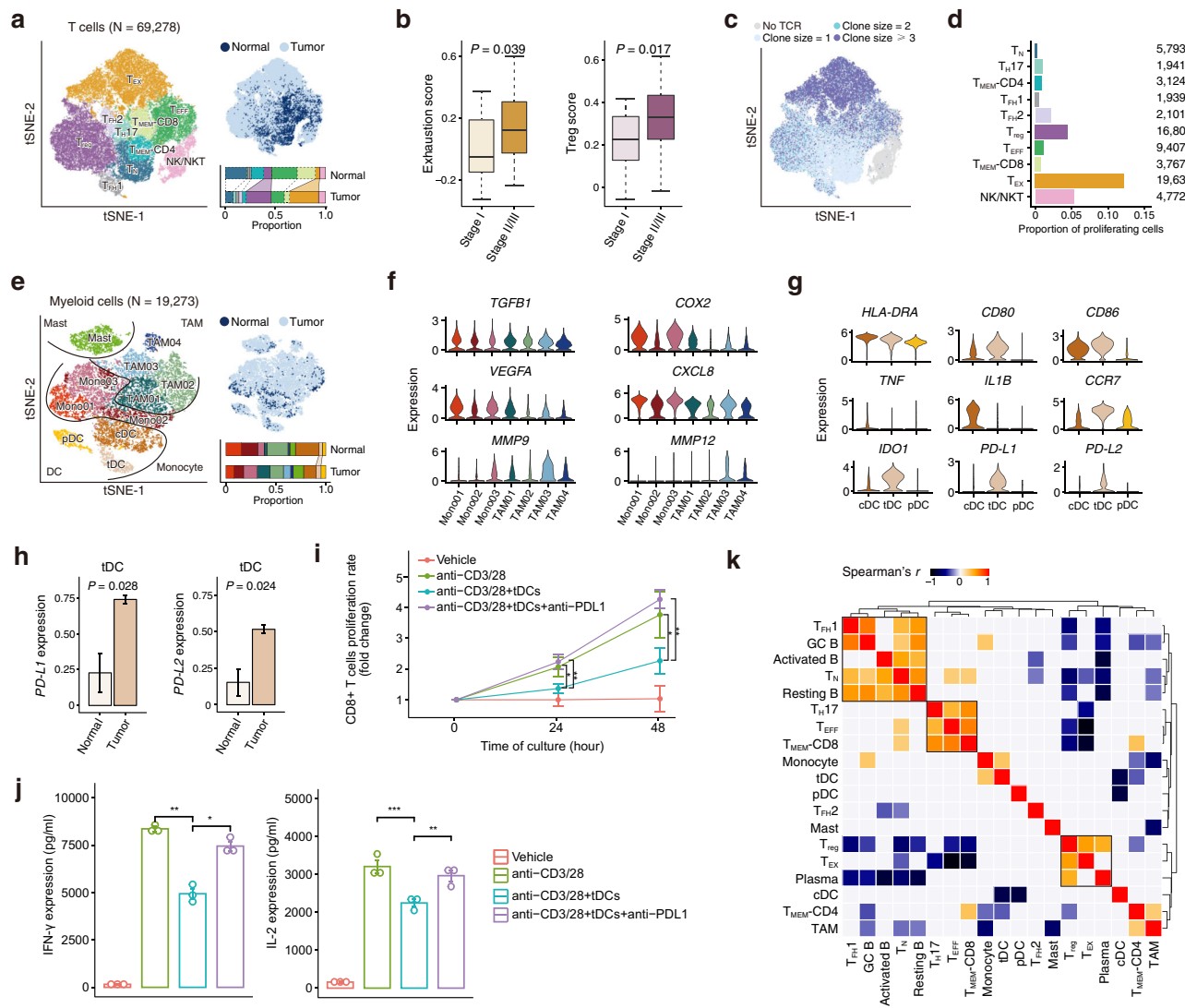

**Fig. 4 Characterization of immunosuppressive ESCC tumor microenvironment. a** tSNE plots of 69,278 T cells, colored by cell type (*left*) and tissue type (*top right*). The proportions of cell types in normal and tumor tissues are shown on the bottom right (colors as in the left panel). See Fig. 1b legend for the line indications. **b** Boxplots showing exhaustion score (*left*) and Treg score (*right*) of T cells are differentially distributed in patients at stage I (N = 16) and stage II/III (N = 44). P values of two-sided Wilcoxon test are shown. Boxplots show the median (central line), the 25–75% interquartile range (IQR) (box limits), the ±1.5 times IQR (Tukey whiskers), and all data points, among which the lowest and the highest points indicate minimal and maximal values, respectively. **c** tSNE plot of T cells colored by clone size. **d** Bar plot showing fractions of proliferating cells in each T cell subtype. The cell numbers of T cell subtypes are labeled on the right. **e** tSNE plots of 19,273 myeloid cells, colored by cell type (*left*) and tissue type (*top right*). The proportions of cell types in normal and tumor tissues are shown on the bottom right (colors as in the left panel). See Fig. 1b legend for the line indications. **f** Violin plots showing the expression distribution of selected genes involved in immune suppression and angiogenesis in the monocyte and TAM clusters. **g** Violin plots showing the expression distribution of selected genes involved in DC maturation and immunosuppression in the three DC clusters. **h** Boxplots showing the expression levels of *PD-L1* and *PD-L2* in tDC are higher in tumor tissues (N = 60) compare with normal tissues (N = 4; two-sided Student's t-test). The data presented as mean ± SEM. **i, j** 100,000 autologous CD8+ T cells isolated from human patients' peripheral blood and tDCs were co-cultured in a 96-well plate at a ratio of 10:1 with Human T-Activator CD3/CD28. Anti-PD-L1 was administered at 100 μg/ml. T cell number was counted by hemocytometer at day 1 and day 2 (**i**). IL-2 and IFN-γ levels of the coculture supernatant at day 2 were measured by BD Cytometric Bead Array (CBA) kit (**j**). In (**i, j**), P values are derived from two-sided Wilcoxon test; *P < 0.05, **P < 0.01, ***P < 0.001. The data presented as mean ± SD and mean ± SEM in (**i**) and (**j**), respectively. Data in (**i, j**) are representative of three independent experiments for each group. **k** Clustered heatmap showing two-sided Spearman correlation coefficients between proportions of immune cell subtypes in tumor samples. P < 0.05 is used as the cutoff value.

were more canonical as they expressed *CXCR5* that helps T$_{FH}$ cells localize to B cell follicles and promote the differentiation and maturation of B cells. We found that ESCC tumors were rich in T$_{reg}$ and T$_{EX}$ cells but poor in T$_N$, T$_{MEM}$, and T$_{EFF}$ cells compared with adjacent normal tissues, indicating an immunosuppressive status in the TME (Fig. 4a; Supplementary Fig. 3c; Supplementary Data 3 and 4). Stage II/III tumors had fewer T$_H$17 and T$_{FH}$1 cells but more T$_{EX}$ cells than stage I tumors (Supplementary Fig. 3d).

We calculated the cytotoxicity score and exhaustion score to quantify properties of CD8+ T cells (Supplementary Fig. 3e, f; see "Methods") and found that T$_{EX}$ cells had the highest cytotoxicity and exhaustion scores. Most of T$_{EX}$ cells were likely tumor-reactive T cells because of high levels of CD39 (*ENTPD1*) and CD103 (*ITGAE*) and very low level of *KLRG1* (Supplementary Fig. 3g)[38–41]. We also generated a Treg score to quantify the activity of CD4+ T cells (Supplementary Fig. 3h). We found stage

II/III ESCC tumors had significantly elevated exhaustion and Treg scores compared with stage I tumors, indicating that the immunosuppressive status in the ESCC TME is worse with tumor progression (Fig. 4b).

We profiled TCR repertoires to analyze the lineage structure of T cells in ESCC and found that the TCR clonotype compositions and proliferative cell proportions were highly diverse across different T cell subtypes (Fig. 4c, d; Supplementary Fig. 3i). The $T_{EX}$ cells had the highest level of clonal expansion (clone size ≥3 cells; Supplementary Fig. 3i) and the highest proportion of proliferation (nearly 10-fold greater than proliferative $T_{EFF}$ cells) (Fig. 4d). By measuring the proportion of clonotypes in a given T cell subtype (primary phenotype) shared with another T cell subtype (secondary phenotype), we found that CD8+ cell subtypes had a high degree of clonal sharing and extensive transitions existed among $T_{EFF}$, $T_{EX}$, and $T_{MEM}$ subtypes (Supplementary Fig. 3j). CD4+ T cell subtypes, except for $T_{FH}1$ and $T_{FH}2$ cells, had a low degree of clonal sharing and null sharing with $T_{reg}$ cells. These results indicate that $T_{EX}$ cells were highly proliferative and dynamic in the immunosuppressive TME throughout ESCC progression. Using the scTCR-seq and scRNA-seq data from 4 patients with matched tumor and adjacent normal samples, we found 23.9% of the T-cell clones in tumors were shared with normal samples, suggesting the parallel expansion of $T_{EFF}$ and $T_{MEM}$-CD8 in these sites (Supplementary Data 7)[42].

We also profiled B cells ($N = 22,477$) in both ESCC tumor and adjacent normal tissues and generated 5 subtypes designated as resting B, activated B, germinal center B (GC B1/B2) and plasma cells (Supplementary Fig. 4a, b; Supplementary Data 6b). High *MKI67* and *TOP2A* expression in GC B2 cells suggested that these cells were active in proliferation (Supplementary Fig. 4b). The presence of GC B, FRCs and $T_{FH}$ cells indicates the existence of tertiary lymphoid structures (TLSs) which plays an antitumor role in the otherwise immunosuppressive TME[43]. These B cell subtypes provided key snapshots of B cells activation, proliferation, and differentiation in ESCC tumors.

**Identification of tDC as a major player in the immunosuppressive ESCC microenvironment.** Since tumor-infiltrating myeloid cells (TIMs) have been shown to be fundamental in regulating both innate and adaptive immune responses and facilitating tumor angiogenesis, invasion and metastasis[44,45], we re-clustered all myeloid cells ($N = 19,273$) to look at their status and potential roles in ESCC. We found that TIMs in ESCC had 11 subtypes comprising 4 categories: monocytes (Mono01-03), tumor-associated macrophages (TAM01-04), mast cells (Mast) and dendritic cells (DC) (Fig. 4e; Supplementary Fig. 4c; Supplementary Data 6c). DC had three distinct subtypes with divergent roles: conventional DC (cDC), tolerogenic DC (tDC) and plasmacytoid DC (pDC). We found that monocytes and TAMs expressed not only genes associated with immunosuppression (e.g., *TGFB1* and *COX2*), but also genes involved in angiogenesis (e.g., *VEGFA*, *CXCL8*, *MMP9* and *MMP12*) (Fig. 4f). *VEGFA* was upregulated in monocytes, while *MMPs* were mainly expressed in TAMs.

Mature cDC had high expression of MHC class II molecules (e.g., *HLA-DRA*), costimulatory factors (e.g., *CD80* and *CD86*), and proinflammatory cytokines (e.g., *TNF* and *IL1B*) that may activate T cells and other antitumor immunity (Fig. 4g). Similar to mature cDC, tDC also had high expression of MHC class II molecules and costimulatory factors, but null expression of the proinflammatory cytokines *TNF* and *IL1B*, suggesting semi-mature phenotype of these cells. In addition, we found that tDC expressed the highest level of *CCR7*, which essentially contributes

to both immunity and immunotolerance[46–48]; pDC specifically expressed *ICOSLG*, a gene involved in the expansion of $T_{reg}$ cells (Supplementary Fig. 4d). Furthermore, among all immune cell subtypes, tDC had the highest expression of the immune checkpoint genes (*IDO1*, *PD-L1* and *PD-L2*) (Fig. 4g; Supplementary Fig. 4d–f). We also found that ESCC tumors were rich in tDC and the expression of *PD-L1/L2* in tDC was significantly higher in tumors compared with adjacent normal tissues (Fig. 4h; Supplementary Fig. 4g; Supplementary Data 3 and 4). Ligand–receptor interaction analysis showed that tDC had stronger interactions with multiple subtypes of T cells than that of cDC, pDC, or TAM in terms of immunosuppression (Supplementary Fig. 4h). Coculture experiment of tDC isolated from ESCC tissue of one patient and autologous CD8+ T cells isolated from human peripheral blood sample with the stimulation by anti-CD3/28 showed that tDCs significantly inhibited CD8+ T cell proliferation (Fig. 4i). Furthermore, CD8+ T cells co-cultured with tDCs had diminished production of IL-2 and IFN-γ. We found that when anti-PD-L1 antibody was included in the coculture, the reduced CD8+ T cell proliferation and suppressed IL-2/IFN-γ production were rescued, suggesting that the suppressing effect of tDC on CD8+ T cell proliferation and activation is likely mediated through PD1 and PD-L1 interaction (Fig. 4i, j). Collectively, these results suggest that tDCs among TIMs play a crucial role in the immunosuppressive ESCC TME.

**Identification of immune cell interwoven entities in the ESCC microenvironment.** We then investigated the interactions and associations among different types of immune cells in tumor samples. Cells were clustered into multiple groups with different association patterns consistent with their related functions (Fig. 4k). We found that $T_N$, $T_{FH}1$, and B cells (activated, resting, and GC B cells) as well as $T_H17$, $T_{EFF}$, and $T_{MEM}$-CD8 cells were aggregated into two major groups that contribute to immune responses. We also identified an immunosuppressive group consisting of $T_{reg}$, $T_{EX}$ and plasma cells and found negative correlations between suppressive and effective immune cell subtypes. Furthermore, we found that the proportions of CXCR5+ $T_{FH}1$ and GC B cells were highly correlated ($r = 0.66$, $P = 2.4 \times 10^{-9}$; Fig. 4k; Supplementary Fig. 4i), which is in line with the known fact that these two cell types are directly interactive in the germinal center[49]. Together, these results demonstrate a broad interaction among these immune cell subtypes that may play an important role in the construction of the immunosuppressive ESCC microenvironment.

**Characterization of specific trajectories for fibroblast and pericyte differentiation.** Fibroblasts and pericytes that can differentiate into myofibroblasts via pericyte-fibroblast transition are two important components of TME that may facilitate tumor invasion and metastasis[23,50,51]. Therefore, we profiled fibroblasts and pericytes to investigate the complexity and dynamic relationship of these cells in the ESCC TME. We found that they can be clustered into 9 subtypes as normal mucosa fibroblasts (NMF), normal activated fibroblasts (NAF1/2), cancer-associated fibroblasts (CAF1–4), pericytes and vascular smooth muscle cells (VSMC) (Fig. 5a; Supplementary Fig. 5a, b; Supplementary Data 6d). These fibroblast subtypes had distinct patterns of marker gene expression and pathway activities as shown in Fig. 5b and Supplementary Fig. 5c. NMF expressed a panel of genes encoding the protease inhibitors (e.g., *SLPI* and *PI16*) and genes involved in the complement, coagulation, peroxisome, and apoptosis pathways. NAF1/2 expressed the wound healing response-related genes (e.g., *IGF1*, *C7*, and *APOD*). CAF1 and CAF2 expressed proinflammatory chemokines (e.g., *CXCL1* and

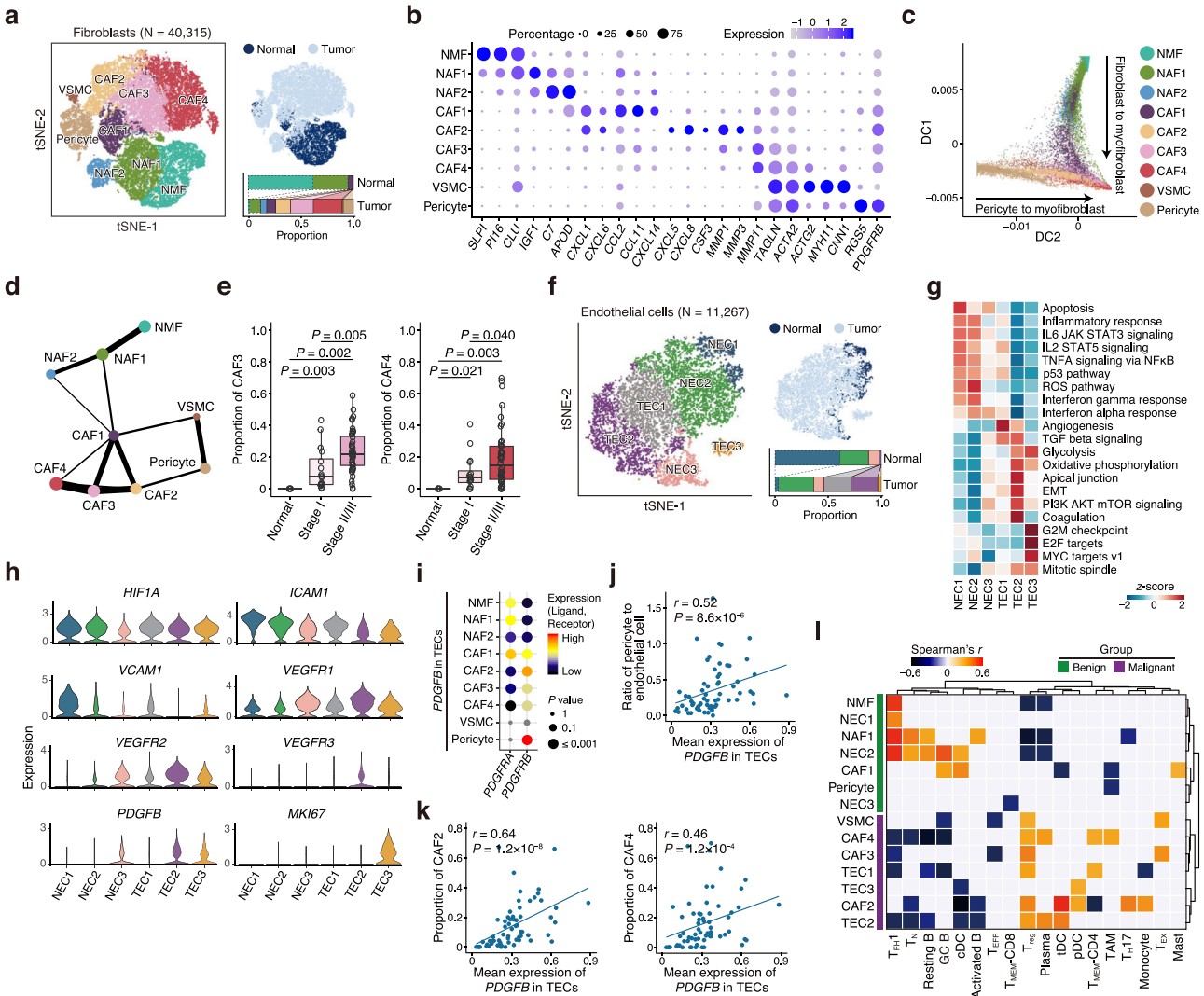

**Fig. 5 Characterization of specific trajectories for fibroblast and pericyte differentiation. a** tSNE plots of 40,315 fibroblasts, colored by cell type (*left*) and tissue type (*top right*). The proportions of cell types in normal and tumor tissues are shown on the bottom right. See Fig. 1b legend for the line indications. **b** Dotplot showing expression status of selected fibroblast markers in each cell cluster. **c** Diffusion map of fibroblasts. Dot represents single-cell colored by cluster. The two main differentiation branches are indicated with solid arrows. **d** PAGA analysis of fibroblast subtypes. Line thickness corresponds to the level of connectivity. **e** Boxplots showing proportion of CAF3 (*left*) and CAF4 (*right*) in normal tissues ($N = 4$), stage I ($N = 16$) and stage II/III ($N = 44$) tumors (two-sided Wilcoxon test). Boxplots show the median (central line), the 25–75% interquartile range (IQR) (box limits), the ±1.5 times IQR (Tukey whiskers), and all data points, among which the lowest and the highest points indicate minimal and maximal values, respectively. **f** tSNE plots of 11,267 endothelial cells, colored by cell type (*left*) and tissue type (*top right*). The proportions of cell types in normal and tumor tissues are shown on the bottom right. See Fig. 1b legend for the line indications. **g** Heatmap showing differences in pathway activities scored per cell by GSVA between different endothelial clusters. **h** Violin plots showing the expression distribution of selected genes in the endothelial cell clusters. **i** Dotplot showing selected ligand–receptor interactions. The means of the average expression level of *PDGFB* in TECs and interacting receptor, *PDGFRA* and *PDGFRB*, in fibroblast subtypes are indicated by color. **j** Spearman's correlation between the mean expression of *PDGFB* in TECs and the ratio of pericyte to endothelial cell in all samples. **k** Spearman's correlations between the mean expression of *PDGFB* in TECs and the proportion of CAF2 (*left*) and CAF4 (*right*) in all samples, respectively. **l** Clustered heatmap showing two-sided Spearman correlation coefficients between proportions of immune and nonimmune cell subtypes in tumor samples. $P < 0.05$ is used as the cutoff value.

*CXCL6*) and other cytokines involved in recruiting immune cells showing an activated inflammatory status. Interestingly, while CAF1 expressed high levels of *CCL2/11* and *CXCL14*, CAF2 cells expressed activated-pericyte-specific cytokines *CXCL5/8* and *CSF3*[52] and had the most significant upregulation of genes in inflammatory response pathway among all fibroblast subtypes. CAF3 showed an expression pattern resembling CAF4 (i.e., myofibroblasts) and expressed lower levels of myofibroblasts' hallmark genes (e.g., *TAGLN, ACTA2,* and *ACTG2*) than CAF4. Both CAF3 and CAF4 had upregulation of apical junction and EMT pathways. Among all fibroblast subtypes, CAF4 had the

highest activities of glycolysis and angiogenesis, two hallmarks of cancer. Furthermore, we found that each subtype had the distinct collagen repertoire; for example, NMF had lower expression of all collagen genes than CAFs, while CAF4 expressed the highest levels of *COL1A1/2, COL3A1, COL5A1/2,* and *COL6A1/2/3* (Supplementary Fig. 5d).

Diffusion map and principal component analysis showed two distinct developmental trajectories of fibroblasts (Fig. 5c; Supplementary Fig. 5e; see "Methods"). In one branch, we detected a sequential continuum of phenotypes from NMF to NAF1/2, CAF1, CAF3 and CAF4, but in another branch, we found that

pericytes, CAF2 and CAF4 formed a continuous and sequential path. We inferred the connectivity structures between the subtypes using partition-based graph abstraction (PAGA) analysis and found that CAF1 were connected to NAFs and CAF4, while CAF2 were the linkage between pericytes and CAF4, which was compatible with the results of diffusion map analysis (Fig. 5d). We also found that TGF-β pathway that is pivotal in the fibroblast-to-myofibroblast transition[53] was activated in CAF1/2/3 (Supplementary Fig. 5c). These results suggest that both normal fibroblasts and pericytes may differentiate into myofibroblasts.

Interestingly, we found that the proportions of NMF and NAF1 were more prevalent in normal tissues than tumor tissues, while CAF2/3/4, VSMC, and pericytes were predominant in tumor tissues compared with normal tissues (Fig. 5a; Supplementary Fig. 5f; Supplementary Data 3 and 4). Moreover, we found that stage II/III ESCC tumors had more CAF3 and CAF4 but fewer CAF1 than stage I tumors (Fig. 5e; Supplementary Fig. 5g). These results indicate an accumulation of CAFs, especially the myofibroblasts, in the ESCC microenvironment that might play crucial roles in tumor progression.

**Identification of high *PDGFB* level in TECs and its role in pericyte-myofibroblast transition.** Since blood vessels and endothelial cells (EC) often exhibit aberrant phenotypes and functions in the TME, which might restrict them to respond to anti-angiogenic therapy, we therefore addressed these issues. We clustered EC (N = 11,267) into 6 subtypes including 3 normal EC (NEC1–3) and 3 tumor EC (TEC1–3) (Fig. 5f; Supplementary Data 6e). NEC1/2 expressed vein endothelial markers (e.g., *CPE* and *ACKR1*) while NEC3 expressed artery markers (e.g., *SEMA3G* and *GJA5*)[54] (Supplementary Fig. 5h). While NEC1 were prevalent in normal tissues, TECs were exclusively present in ESCC tumors and TEC2/3 had increased expression of *ANGPT2* that disrupts pericyte–EC interactions to enable angiogenesis[23] (Fig. 5f, Supplementary Fig. 5h, i; Supplementary Data 3 and 4). We found that TECs had specifically activated pathways involved in cell proliferation, angiogenesis, TGF-β signaling, EMT and energy metabolism (Fig. 5g). Notably, TECs had lower expression levels of genes associated with antigen presentation (e.g., *CD74* and *HLA-DQA1*) and cell adhesion (e.g., *ICAM1* and *VCAM1*) than NECs, which has been known to impede immune cell infiltration[55], but had higher expression of molecules associated with angiogenesis (e.g., *VEGFR1/2/3* and *PDGFB*) than NECs (Fig. 5h).

Since PDGFB-PDGFRB signaling pathway play a pivotal role in pericyte-myofibroblast transition and PDGFB is mainly secreted by endothelial cells[51], we analyzed its role in the interaction of TECs and fibroblasts especially pericytes. We found a strong interaction between TECs and pericytes based on the ligand–receptor interaction analysis (Fig. 5i). The ratio of pericytes to ECs was positively correlated with the levels of *PDGFB* in TECs ($r = 0.52$, $P = 8.6 \times 10^{-6}$; Fig. 5j), in line with the recruitment role of *PDGFB* for pericytes. In addition, the expression levels of *PDGFB* in TECs were correlated with the proportions of CAF2 ($r = 0.64$, $P = 1.2 \times 10^{-8}$) and CAF4 ($r = 0.46$, $P = 1.2 \times 10^{-4}$; Fig. 5k). Furthermore, SCENIC analysis showed that several ETS family transcription factors (EHF, ELF3, and ETS2) downstream of MAPK were activated in CAF2 that can be induced by the PDGFB-PDGFRB pathway (Supplementary Fig. 5j)[56]. These results suggest that TECs in the ESCC TME may promote tumor progression by facilitating tumor angiogenesis via inducing pericytes to myofibroblasts transition.

**Interactions among immune and nonimmune stromal cells in the ESCC microenvironment.** We next explored the associations

between immune and nonimmune stromal cells by clustering the related cell subtypes according to their association patterns, resulting in the benign and malignant groups (Fig. 5l; Supplementary Fig. 5k–m; Supplementary Data 3; see "Methods"). The benign group comprised the NMF, NAF1, CAF1, and NECs subtypes, while the malignant group comprised CAF2−4, TECs, and VSMC. We observed strong associations between benign group and effective immune cells, including resting B, GC B, $T_N$, $T_{FH}1$, cDC and activated B cells; however, the malignant group tended to correlate with suppressive immune cells, such as $T_{reg}$, $T_{EX}$, TAM, tDC, and pDC. $T_{reg}$ cell proportions were positively correlated with CAF2−4 and TEC1/2 proportions but negatively correlated with NMF, NAF1 and NEC2 proportions. $T_{EX}$ cell proportions were positively correlated with VSMC and CAF3 proportions. In addition, the proportions of $T_H17$ cells that produce IL-17, a cytokine associated with pericyte activation[52], were correlated with the proportions of CAF2 ($r = 0.36$, $P = 0.004$; Supplementary Fig. 5n), which is in line with the developmental trajectories of pericytes, suggesting that CAF2 may be activated pericytes. These extensive interactions among immune and nonimmune stromal cells might shape an immunosuppressive ESCC TME that promoted tumor progression. We also found that some clinical characteristics of patients such as sex, smoking status, and drinking status were significantly associated with the proportions of TME cell types (Supplementary Fig. 5o).

**Associations of epithelial expression programs and genomic alterations with TME compositions.** We next wanted to look at the interactions between malignant epithelial cells and other cells in the ESCC TME (Fig. 6a; see "Methods"). We found that ESCC tumors with higher Mucosal program scores had more $T_N$, $T_{FH}1$, NK/NKT, GC B, and cDC but fewer $T_{EX}$, plasma, tDC, and pDC. Such ESCC tumors also possessed more NEC2 but fewer CAF2−4. ESCC tumors with higher AP program scores exhibited more $T_{EFF}$ cells, resting B cells and NEC2 but fewer CAF4 and TEC2. However, ESCC tumors with higher Cycling program scores had fewer $T_{MEM}$-CD8 cells, NMF, NAF1/2, CAF1, and NEC1/2 but more CAF2/3 and TEC2/3 as well as higher cytotoxicity, exhaustion, and Treg scores. ESCC tumors with high Epi1 scores had reduced infiltrating effective immune cells ($T_{FH}1$ and GC B) and benign stromal cells (NMF, NAF1/2, and NEC1/2). Conversely, such ESCC tumors contained more malignant CAF3/4 and TEC1 and had higher cytotoxicity and Treg scores.

We next sought to know whether the genomic alterations may affect the ESCC ecosystem. We detected 11 putative driver genes mutated in ESCC by whole-exome sequencing (WES) and among them the most frequently mutated genes were *TP53* (89.1%) and *NOTCH1* (28.3%) (Supplementary Fig. 6a). By analyzing the differences in epithelial expression program scores and TME cell-type proportions associated with *TP53* or *NOTCH1* mutation status, we found that ESCC tumors with the mutant *TP53* had higher Cycling and Epi1 program scores and higher immune suppression levels than tumors with the wild-type *TP53* (Fig. 6b; Supplementary Fig. 6b). The *TP53* mutations were also associated with higher cytotoxicity and Treg score and increased TAMs but decreased $T_N$ cell and monocyte infiltration. In contrast, ESCC tumors with *NOTCH1* mutations had lower Epi1 and Epi2 program scores but higher Mes scores, reflecting the role of mutant *NOTCH1* signaling in epithelial differentiation (Fig. 6b; Supplementary Fig. 6c). Tumors with *NOTCH1* mutations also had increased $T_{EFF}$ cell infiltration than tumors without *NOTCH1* mutations, indicating that the mutations might have produced neoantigens. By integrative analyzing the mutations of *TP53* and *NOTCH1*, we found sole mutation of *TP53* was associated with a

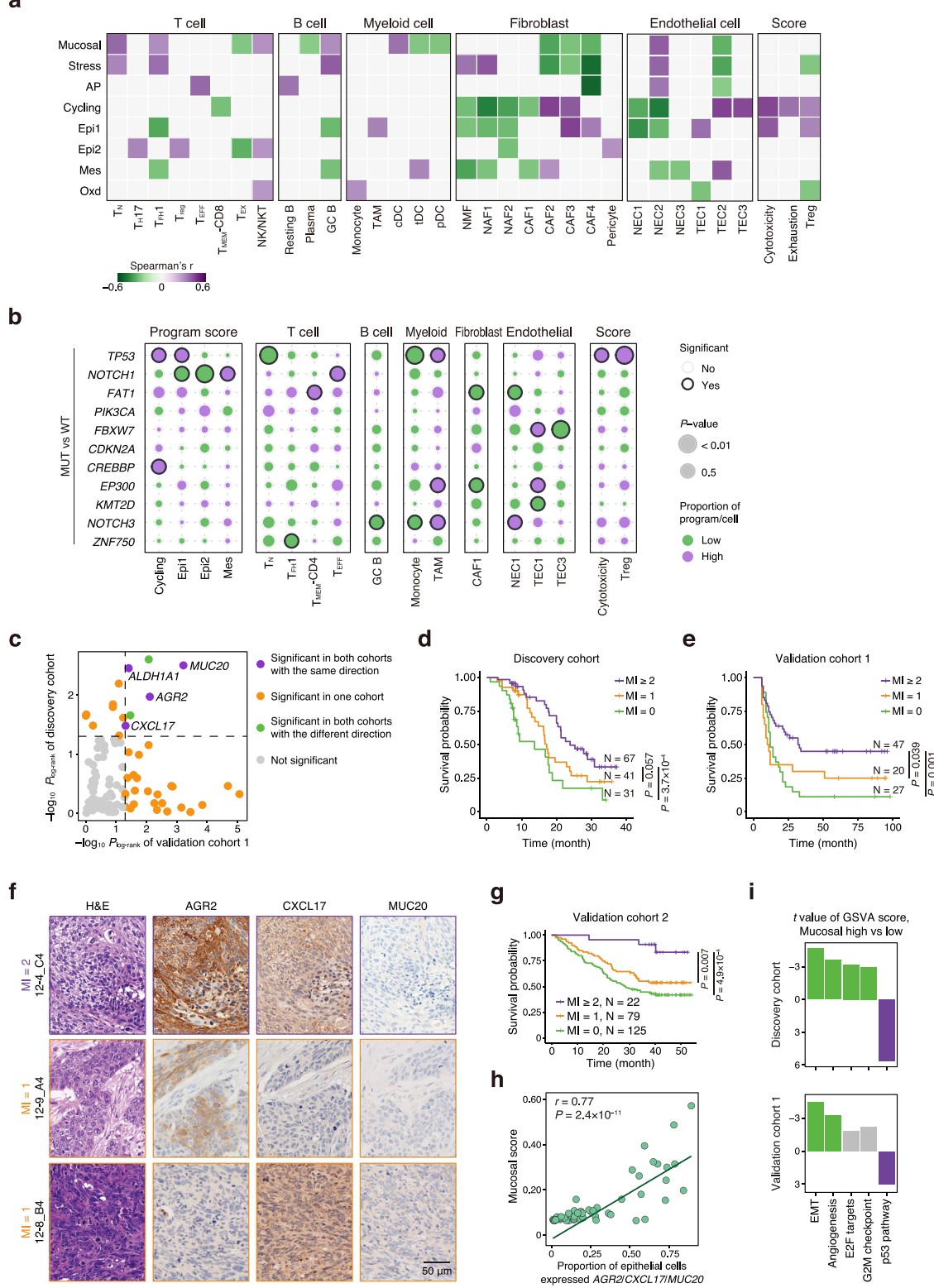

reduced proportion of $T_N$, $T_{FH}1$, GC B, resting B, monocyte, and NEC2 and increased proportion of plasma, TAM and TEC1 as well as higher cytotoxicity and Treg score (Supplementary Fig. 6d). For tumors with *TP53* mutation, the mutation of *NOTCH1* presented higher infiltration of $T_{EFF}$ cells but reduced Epi1 and Epi2 scores than wild-type *NOTCH1*. The mutational signatures were also associated with the heterogeneity level of malignant cells and the TME compositions (Supplementary

Fig. 6e, f). These results suggest that somatic mutations of some driver genes in esophageal epithelial cells may drive the alterations of epithelial expression programs and the TME compositions that facilitate tumor progression.

**Correlation of gene expression levels in the Mucosal program with ESCC survival in patients**. Since epithelial expression

**Fig. 6 Correlation of gene expression levels in the Mucosal program with ESCC survival. a** Heatmap showing two-sided Spearman correlation coefficients between program scores and proportions of TME compositions as well as T cell function associated scores (hereafter refers to as T cell scores) in tumor samples. $P < 0.05$ is used as the cutoff value. **b** Bubble plot showing relationships between mutation status of WES-derived frequently mutated genes and program scores, proportions of TME compositions as well as T cell scores. Compositions that significantly increase or decrease in mutated tumors (two-sided Wilcoxon test, $P < 0.05$) are indicated by purple or green, respectively. Circle size represents $P$ value. MUT, mutant; WT, wild type. **c** Associations between program gene expression levels and ESCC survival in Discovery cohort ($N = 139$) and Validation cohort 1 ($N = 94$). The significances of genes in two cohorts are indicated by color. $P_{log-rank} < 0.05$ (two-sided) is used as the cutoff value. **d** Kaplan–Meier (KM) curves of Discovery cohort ($N = 139$) grouped by the mucosal index (MI). **e** KM curves of Validation cohort 1 ($N = 94$) grouped by the MI. **f** Representative images of IHC of the proteins produced by the 3 Mucosal genes in Validation cohort 2. Shown are samples with MI 1 (12-9_A4 and 12-8_B4) and MI 2 (12-4_C4). Scale bar, 50 μm. **g** KM curves of Validation cohort 2 ($N = 226$) grouped by the MI. **h** The relationships between the proportion of epithelial cells expressed *AGR2*, *CXCL17* or *MUC20* (total expression >0) and Mucosal score in tumor samples. Two-sided Spearman correlation coefficient and $P$ value are indicated. **i** Differences in selected pathway activities by GSVA between tumors with high or low expression level of Mucosal genes in Discovery cohort and Validation cohort 1, respectively. Shown are $t$ values from the linear model. $P$ values in (**d**, **e**, **g**) are derived from log-rank tests.

programs may reflect the genomic alteration information and drive remodeling of ESCC TME compositions, we therefore analyzed the association between epithelial expression programs and the disease outcome. We found that the expression levels of the identified 14 genes were significantly associated with survival time in ESCC patients (Discovery cohort, $N = 139$; all $P_{log-rank} < 0.05$; Fig. 6c, Supplementary Data 1b and 8). Among the 14 associated genes, the *AGR2*, *CXCL17* and *MUC20* in the Mucosal program were also significantly associated with survival time in a validation cohort of 94 ESCC patients (Validation cohort 1; all $P_{log-rank} < 0.05$; Supplementary Fig. 7a, b; Supplementary Data 1c and 8). *AGR2* encodes an endoplasmic reticulum protein essential for intestinal mucus production that mediates fundamental innate immunity[57]. *CXCL17* produces a mucosa-associated chemokine that supports innate immunity and the homeostasis and integrity of the mucosa[58]. *MUC20* encodes a mucin protein that is part of an insoluble mucous barrier[59]. These three genes are specifically expressed in epithelial cells so that can be measured by bulk-tissue RNA-seq (Supplementary Fig. 7c). We further categorized the expression level of these 3 genes as high (>median) or low (≤median) and integrated them as the Mucosal Index (MI) for further analysis. We found that in Discovery cohort and Validation cohort 1, patients with high MI (MI ≥ 2) had significantly longer median survival time than patients with medium MI (MI = 1) ($P = 0.057$ and $P = 0.039$, respectively) or low MI (MI = 0) ($P = 3.7 \times 10^{-4}$ and $P = 0.001$, respectively) (Fig. 6d, e). The correlations of the expression levels of 3 genes and ESCC survival time were further validated in another cohort consisting of 226 patients by immunohisto-chemical staining of protein levels in ESCC tumors (Validation cohort 2; Fig. 6f; Supplementary Data 1d) and the results also showed that higher expression levels of their proteins (expressed as H-score) were correlated with significantly longer ESCC survival time (all $P_{log-rank} < 0.05$; Fig. 6g; Supplementary Fig. 7d–f). We found that the proportions of epithelial cells expressing *AGR2/CXCL17/MUC20* were correlated with the Mucosal program scores in scRNA-seq data ($r = 0.77$, $P = 2.4 \times 10^{-11}$; Fig. 6h), suggesting that these three genes could be collectively used to predict the Mucosal program activity.

We further investigated Mucosal program-correlated epithelial aberrations in bulk-tissue RNA-seq data of discovery cohort and validation cohort 1 using GSVA (Fig. 6i). Among hallmark pathways differentially expressed between tumors with the high or low expression level of Mucosal genes, the EMT and angiogenesis pathways were significantly downregulated in Mucosal-high tumors in both cohorts, suggesting suppressed migration and proangiogenic abilities of tumors with high Mucosal activities. Tumor suppressor p53-related pathway was significantly enriched in Mucosal-high tumors in both cohorts, suggesting that Mucosal-high tumors are relatively less malignant.

We also found the E2F targets and G2M checkpoint pathways were downregulated in Discovery cohort, suggesting suppressed cell proliferation of tumors with high Mucosal activity. These results collectively point to a relative less malignant phenotype and better TME of Mucosal-high ESCC (Fig. 6a), which might explain why patients with mucosal-high ESCC had a better prognosis.

## Discussion

In the present study, we have performed scRNA-seq in 208,659 cells from ESCC tumor and their adjacent normal samples collected from 60 patients and deciphered the phenotypes and compositions of the ESCC ecosystem. We found that samples can be reliably classified into two groups according to the different adoptions of shared clusters for epithelial cells in regard to the sample compositions and these differences can be further quantified using PCA-based heterogeneity scores. We have dissected 8 common expression programs from malignant epithelial cells and decomposed the TME compositions into 42 functional subtypes including 26 immune cell subtypes and 16 nonimmune stromal cell subtypes. We have also elucidated the possible interactions between cancer cells and TME cells and the interactions among different cell types in the TME. Moreover, we have dissected the relationships between ESCC ecosystem components and genomic alterations or disease outcomes. These results provide funda-mental information for deeply understanding the mechanism for pathogenesis and progression of ESCC, which in turn may benefit the development of approaches for precision cares of ESCC patients.

The present study has provided several significant findings. Firstly, we have identified an immunosuppression status in ESCC TME featured by a large amount of $T_{EX}$, $T_{reg}$, and myeloid cell infiltration. Specially, we have revealed that in the TME, tDC express the highest levels of immune checkpoint genes (*PD-L1/2* and *IDO1*) that have been shown to induce the T cell anergy and produce $T_{reg}$ cells[60,61]. Our experimental results showed that tDC can suppress the activation of CD8+ T cells, suggesting that they may play an important role in evoking immunosuppression. Given the central role of dendritic cells in controlling immunity homeostasis, measurement of tDC in ESCC tumors might predict whether the tumor is vulnerable to immunotherapy and targeting both tDC and T cells may be necessary in immunotherapy of ESCC, as reported in prostate cancer for a synergistic effect[62]. Furthermore, we did not identify the existence of recently reported MR1-restricted T cells, which have been reported to recognize a broad range of cancer cells[63]. This negative result might further support that ESCC tumors have the immunosuppressive TME.

Secondly, we have deciphered the complicated compositions of nonimmune stromal cells and their dynamic transition in the

ESCC TME. We have discovered two hidden intermediate phenotypes of fibroblasts (CAF1 and CAF2) jointly manifesting two developmental trajectories among them. These transitionary fibroblast subtypes are activated and produce various proinflammatory cytokines that play important roles in ESCC progression. We have also found that the characteristic high expression of *PDGFB* in TECs is related to the proportions of pericytes, CAF2, and myofibroblasts in the ESCC TME, likely indicating that TECs play a part in reshaping the TME. These findings imply that targeting PDGFB signaling pathway may inhibit angiogenesis via blocking TECs to recruit pericytes and repressing the pericyte-myofibroblast transition[51,64,65], which offers a new therapeutic option for cancer treatment. One limitation is that the differentiation trajectories for fibroblasts and pericytes and the role of *PDGFB* in ESCC TME need to be verified by further functional studies.

Thirdly, we have identified several different cellular interactions in the TME, such as the interaction between $T_{FH}1$ and GC B cells, which produces an antitumor immune environment by the formation of TLS, the interaction between TEC2 and $T_N$ cells, which impedes infiltration of immune cells to tumor tissues, and the interaction between $T_H17$ cells and CAF2 mediated by cytokines secreted from $T_H17$ cells, which enhances CAF2 cell activity. CAF2 are the intermediate activated phenotype in the pericyte-myofibroblast transition and $T_H17$ cells are implicated in activation of pericytes[52], suggesting that $T_H17$ cells may play an important role in pericyte-myofibroblast transition. We have found that the amount of cell subtypes that are phenotypically or functionally relevant is highly correlated and these cells can be grouped as malignant nonimmune stromal cells (e.g., CAFs and TECs) and suppressive immune cells (e.g., $T_{reg}$, $T_{EX}$, and TAM). These results suggest that cells in the ESCC TME are interwoven together for promoting tumor survival and development. Interestingly, we have found an extensive cooperation between active cancerous epithelial cells, exhibited by the expression programs, and malignant nonimmune cells or suppressive immune cells. The proliferative tumors had high amount of CAFs and TECs in their TME, providing aberrant extracellular matrix, cytokines and vascular niche for tumor cells to be progressive and invasive[19]. These results present a high-resolution picture of entire and detailed cellular interactions in the ESCC tumor ecosystem, which would be significant in developing new therapeutic strategies.

Furthermore, we have also linked the single-cell transcriptomes to the genomic alterations detected by bulk-tissue WES. The results have shown that the mutations in *NOTCH1* are associated with decreased expression of the Epi1/2 programs but elevated expression of the Mes program in epithelial cells, indicating that the mutations may impair the differentiation of epithelial cells, a well-known hallmark of cancer. However, ESCC having the *NOTCH1* mutations seem to have more infiltrated $T_{EFF}$ cells, suggesting that tumors with the mutations have more effective immunity. In contrast, the *TP53* mutations confer ESCC to be more proliferative and immunosuppressive and have an effect on the expression of the Epi1 program opposite to *NOTCH1* mutations. It has been shown that the *NOTCH1* mutations are less frequent in ESCC tumors than in aged normal esophageal tissues[8,9]. This phenomenon can be explained by scRNA-seq results showing that tumors with the *TP53* mutations have higher transcriptomic expressions related to proliferation and immunosuppression than tumors with the *NOTCH1* mutations. The proliferative advantage of cancer cells may result in accumulation of *TP53* mutations in ESCC, whereas the differentiation-related *NOTCH1* mutations are accumulated in aged noncancerous esophageal epithelium.

Most importantly, we have found the expression levels of the mucosal immunity-like (Mucosal) program are significantly associated with ESCC survival time in patients. Because the high expression levels of the Mucosal program in tumor are strongly associated with the amount of infiltrating effective immune compositions such as $T_{FH}1$, GC B, and cDC, one may expect that patients with high expression of the Mucosal program would have higher antitumor immunity and thus have better prognosis. Notably, GC B cells are the major cell type in TLS, which have been shown to be able to promote immunotherapy response in several types of malignancies[66–68]. We have identified that within the program the expression levels of *CXCL17*, *AGR2* and *MUC20* are the best markers associated with ESCC survival. Although the underlying mechanisms for *AGR2* and *MUC20* are currently not evident and warrant further investigation, *CXCL17* has been reported as an antitumor factor since it can recruit dendritic cells into tumors, which may enhance antitumor immunity[58]. These results may be clinically relevant in precision and individualized care of ESCC patients. Furthermore, the production of genes in the Mucosal program, such as *CXCL17*, might be useful in treatment of cold ESCC tumors.

In summary, the present study provides an atlas of the ESCC ecosystem based on the large-scale single-cell RNA-seq results. We have depicted a comprehensive picture of interactive network among cancerous epithelial cells, stromal cells, and various infiltrating immune cells in the ESCC tumors and their TME. We have also linked the transcriptome in cancerous epithelial cells to their somatic genome alterations and identified several significant markers associated with patients' survival. These results deepen and extend our understanding of the complexity of ESCC tumors.

## Methods

**Human biospecimen collection**. In this study, we carried out integrative analysis in 4 independent cohorts consisting of 519 patients with ESCC. The cohort for scRNA-seq analysis was comprised of 60 patients including 44 males and 16 females. Their ages ranged from 40 to 78 with a median of 63.5. Among these patients, 16 patients were diagnosed as stage I, 18 patients as stage II, and 26 patients as stage III. Fresh ESCC tumors, adjacent normal esophagus tissues (at least 5 cm away from tumor site), and peripheral blood samples were collected at the time of surgery in 2018. For survival analysis, the Discovery cohort containing 139 ESCC patients including 100 males and 39 females were recruited between 2015 and 2017, and their ages ranged from 42 to 82 with a median of 65; Validation cohort 1 comprised 94 ESCC patients including 83 males and 11 females were recruited between 2010 and 2014 as described in our previous report[5], and their ages ranged from 43 to 77 with a median of 61; and Validation cohort 2 consisting of 226 ESCC patients including 169 males and 57 females were obtained between 2015 and 2016, and their ages ranged from 44 to 78 with a median of 63. All patients were unrelated Han Chinese hospitalized in the Linzhou Cancer Hospital and Linzhou Esophageal Cancer Hospital (Henan Province, China). ESCC was confirmed by histopathological examination of surgically removed tumors or biopsy specimens. All patients were not treated with chemotherapy or radiotherapy before tumor resection. Biospecimens were collected immediately upon removal from patients and analyzed as detailed below. This study was approved by the Institutional Review Boards of Cancer Hospital, Chinese Academy of Medical Sciences. Informed consent was obtained from each patient, and clinical information was collected from medical records.

**Sample collection and processing for single-cell RNA sequencing**. Fresh ESCC tumors and their adjacent normal tissue were placed in RPMI-1640 medium (Invitrogen) with 20% fetal bovine serum (FBS; GE Healthcare Life Sciences) on ice immediately after surgical resection. Tissue sample processing was completed 10 h after collection. A portion of sample was cryosectioned, hematoxylin-eosin (H&E) stained, and microscopically examined to assure that the tumor sample contains >40% cancer cells and normal tissue contains no cancer cells. Another portion of the same sample was designated for bulk whole-exome sequencing (WES) or whole-genome sequencing (WGS). The remaining tissue was processed for single-cell RNA sequencing (scRNA-seq).

**Preparation of single-cell suspension**. Fresh ESCC tumors and adjacent normal tissues were rinsed with PBS, gently cut into small pieces on ice and digested in RPMI-1640 medium (Invitrogen) containing 2 mg/ml collagenase IV (Gibco) and 0.5 mg/ml hyaluronidase (Sigma Aldrich) for 1 h at 37 °C. The digested cell suspension was subsequently filtered through a 70-μm cell strainer (BD Biosciences) and incubated in 1x red blood cell lysis buffer (BD Biosciences) on ice for 5 min.

The remaining cells were suspended in 50 μl of PBS containing 1% FBS after washing once with the same medium.

**Preparation of single-cell RNA sequencing and T cell receptor sequencing libraries.** Single-cell suspension was stained with CD45-FITC (BD Biosciences, 555482, dilution 1:100) and sorted into immune (CD45+) or nonimmune (CD45−) cells using a FACSAria flow cytometer (BD Biosciences, BD FACSDiva (version 8.0.1); Supplementary Fig. 1a). Sorted cells were examined and counted before sending to 10x Genomics Chips. We targeted for 7000 cells recovered from each channel and used Chromium Single Cell 5′ Reagent Kits (10x Genomics) to prepared whole transcriptome RNA-sequencing libraries. For CD45+ cells, we also used Chromium Single Cell V(D)J Enrichment kit to analyze the T cell receptor (TCR) sequences. All libraries were sequenced on Illumina HiSeq X Ten with 2 × 150 bp paired-end mode.

**Processing of single-cell RNA-sequencing data.** We processed the scRNA-seq data using Cell Ranger Single-Cell Software Suite (10x Genomics, version 2.1.0) with default parameters, aligned to the GRCh38 reference genome and the raw gene expression matrices were generated for each sample. On average, we recovered 1804 CD45− and 1841 CD45+ cells from each sample with sequencing depth of 143,559 and 80,615 reads per cell respectively. The Seurat package (version 2.3.4) was used for quality filtering and downstream analysis. For quality filtering, we removed genes whose expressions were detected in <0.1% of all cells and filtered out cells that had gene counts <500 or mitochondrial RNA content >20%. Genes that had highly variable expression were selected based on average expression and dispersion level thresholds using the FindVariableGenes function with default settings. The normalized expression levels for each gene were further linearly regressed against the total UMI counts and mitochondrial RNA content per cell using the ScaleData function and performed principal component analysis (PCA) with RunPCA. We performed graph-based Louvain clustering on the top 10 principal components (PCs) using FindClusters. The resolution parameter (Res) set to 0.6 and k for the k-nearest neighbor algorithm set to 30 for most clustering analysis if not particularly indicated. Specially, for endothelial cells, the Res = 0.3 and for T cells, the Res = 2.0. The annotated clusters were robust and consistent by using other clustering parameters (PCs = 30, Res = 0.3 or 0.6). The cluster-specific marker genes were identified using the Wilcoxon test implemented in FindAllMarkers function with Bonferroni correction of P values. The cells clusters were manually annotated according to these marker genes. Finally, gene expression and clustering results were visualized on a tSNE plot of the top ten PCs using RunTSNE.

**Annotation of cell types.** Cell types were annotated based on the expression of known markers, i.e., *EPCAM*, *SFN*, *KRT5*, and *KRT14* for epithelial cell; *FN1*, *DCN*, *COL1A1*, *COL1A2*, *COL3A1*, and *COL6A1* for fibroblast; *VWF*, *PECAM1*, *ENG* and *CDH5* for endothelial cell; *RGS5*, *MCAM* and *ACTA2* for pericyte; *CCL21* and *PDPN* for fibroblastic reticular cell; *CD2*, *CD3D*, *CD3E* and *CD3G* for T cell; *CD19*, *CD79A*, *MS4A1*, *JCHAIN* and *MZB1* for B cell and *CD68*, *LYZ*, *CD14*, *IL3RA*, *LAMP3*, *CLEC4C* and *TPSAB1* for myeloid cell. We introduced previously defined epithelial markers, *EPCAM*, *SFN*, and all expressed cytokeratins for CD45− cells[30] to identify epithelial cells. For each CD45− cluster, we calculated the average normalized expression level of the epithelial marker gene set and expressed as epithelial score. The clusters were ordered by epithelial scores and a significant drop between epithelial and non-epithelial clusters was seen. We defined a robust epithelial score threshold as 10 for epithelial clusters. Subclustering of epithelial cells, fibroblasts, endothelial cells, T cells, B cells and myeloid cells was further performed with the same approach as CD45− and CD45+ cells. For epithelial cells, tumor samples (N = 52) with >100 epithelial cells were retained for subclustering. For T-cell specific clustering, we additionally removed genes related to general cell stress, type I Interferon (IFN) response and cell cycle as previously suggested[41]. The contaminating immune cells in CD45− dataset and nonimmune cells in CD45+ dataset were removed prior to clustering on the basis of their expression patterns. T cells with expression of both *CD4* and *CD8A/B* were considered as unresolved T cells, which were not included in downstream analysis.

**Detection of single-cell copy number variations.** Copy number variations (CNV) in each epithelial cell were estimated by expression level from the scRNA-seq results using an approach similar to that described previously[30]. Briefly, we sorted genes according to their genomic location and calculated average gene expression within each chromosome using a sliding window of 100 genes to deduce the CNV. Epithelial cells from normal tissues were used as a normal karyotype reference for the estimation. The numbers of epithelial cells in normal tissues were limited due to the thin epithelium of normal esophagus (about 25 cell layers thick)[69] and vulnerable characteristic of normal epithelial cells[70,71]. However, it did not impact their role as reference for copy number inferences.

Compared to CNV calculated from bulk WGS data, the resolution of CNV inference using scRNA-seq data is limited. The sliding windows of this method require 100 genes which usually span about than 50 Mb in the genome for tumor cells, which hinders the ability to identify small CNV segments. However, for those

large CNVs spanning the whole chromosome or chromosome arm, they were in good concordance with WGS result.

**Comparison of the cell proportions between the adjacent normal tissues and ESCC tumors.** The Wilcoxon test was used to explore the cell-type proportion changes between adjacent normal tissues and ESCC tumors. As the number of tumor tissues (N = 60) greatly exceeded that of normal tissues (N = 4), for significantly changed cell types, we performed down-sampling analysis to remove the influence of this unbalanced sample sizes. We randomly selected 4 samples among all tumor samples without replacement for 1000 times and performed the Wilcoxon test on corresponding cell proportions. We counted the number of tests with P-value less than 0.05 to confirm the cell-type proportion changes under the reduced sample size. For most cell types, we found their differences of cell proportion between adjacent normal tissues and ESCC tumors were robustly validated in more than 50% of the tests. Three cell types, namely $T_{EFF}$, tDC, and NAF1, had less than 50% of the tests with P-value less than 0.05. However, we could still detect the cell-type proportion changes under a moderate significance threshold (P < 0.10) supported by about 50% of the tests. It could be associated with their relative high variance among samples that influence the result of statistical tests. By down-sampling tumor tissues into 8 samples, all the claimed cell types showed significant proportion changes with P < 0.05 in more than 50% of the tests.

**Determination of heterogeneity levels of epithelial cells.** Epithelial cells were subclustered using the Seurat pipeline as described above. To capture major information and reduce the noise, we used PCs instead of original gene expression profiles to measure the intra-tumor and inter-tumor heterogeneity levels. We demonstrated the correlation of intra-tumor and inter-tumor heterogeneity scores calculated using the number of PCs ranging from 10 to 50 (Supplementary Fig. 2c). The intra-tumor heterogeneity scores were highly robust to the number of PCs and inter-tumor heterogeneity scores were robust and stable for PC number exceeding 15. We found that using PCs of 30 gave satisfactory results showing high correlation coefficient with more numbers of PCs (r ≥ 0.98). Accordingly, we adopted 30 PCs for the further analyses. The coordinates of global average and patient average in the PC space were calculated by averaging the PC scores for all epithelial cells or cells from each patient. The intra- or inter-tumor heterogeneity of each patient was defined as the average Euclidean distance between all epithelial cells from the patient and the Patient average or the Global average.

Besides the PCA-based method, we also used cellular correlation coefficients (Pearson) to quantify the heterogeneity scores. Specifically, we performed the correlation analysis on the gene expression matrix of all epithelial cells for each sample and calculated the complement of averaged correlation coefficients as the correlation-based intra-tumor heterogeneity. We also performed the cross-correlation analysis between transcriptome data of cells from individual sample and all other cells and used the complement of averaged correlation coefficients as the correlation-based inter-tumor heterogeneity. We found that they were significantly correlated with the PCA-based intra-tumor (r = 0.71, P = 3.2 × 10⁻⁹) and inter-tumor (r = 0.64, P = 2.9 × 10⁻⁷) heterogeneity scores (Supplementary Fig. 2c). We found that PCA-based scores differentiate better the differences of relative heterogeneity between patients with and without Group 1 cluster. (Supplementary Fig. 2d).

To demonstrate the robustness of the thresholds for Group1 and Group2 clusters and samples (0.75 for cluster and 0.6 for sample), we have calculated the ratio of samples in overlap with the 21 Group 1 cluster for each cluster and sample threshold set. The result shows that under very loose thresholds such as 0.4 for cluster and 0.4 for sample, 70% of the Group 1 samples match to the threshold of 0.75 for cluster and 0.6 for sample, indicating that the Group 1 samples we identified are robust to the choice of thresholds of separation. To validate this analytical method in different sample numbers, we downsampled the original 52 samples to 40% (N = 20), 60% (N = 30) and 80% (N = 40) of samples and found that in each case, the Group 1 samples can be robustly identified.

**Identification of epithelial expression programs.** For each of the 52 ESCC samples that had >100 epithelial cells analyzed, we re-clustered the epithelial cells individually using the Seurat pipeline as described above. In total, we gathered 274 subclusters across the 52 samples and the top 30 marker genes of each subcluster were defined as an expression module for further analysis. The 274 gene modules were further hierarchically clustered based on their expression profiles and 8 epithelial expression programs were identified. The top-scoring genes (N = 30) of each program were manually selected as program genes. These genes were further hierarchically clustered to identify the most significantly correlated gene modules and refine the marker genes for each program. To explore the program expression status of individual cells, we applied a uniform threshold for all programs. For a single cell, if it expressed ≥70% of the genes within a given program, it was considered as a cell having the activated program, namely, program cell.

As the stress program was probably associated with technical factors such as tissue dissociation, we regressed out the expression of key stress markers (*DUSP1*, *EGR1*, *FOSB*, *JUNB*, *FOS*, *BTG2*, and *DNAJB1*) in the normalized expression

matrix for each sample and then performed the program analysis. After removing the influences of stress genes, we still detected the Stress program in epithelial cells but with reduced abundance. All the other programs were retained, and we did not find new programs.

**Analysis of gene set variation**. To functionally describe the epithelial expression programs and non-epithelial cell subtypes, we performed the pathway analysis based on the 50 hallmarks from the MSigDB database (version 6.2)[72] and estimated the pathway activity of individual cells using the gene set variation analysis (GSVA) package (version 1.30.0)[73] with standard settings. To assess the differential activities of pathways between program cells and non-program cells or between different cell subtypes, activity scores were contrasted for each cell group using Limma package (version 3.38.3)[74].

**Analysis of epithelial expression program dependency**. Pairwise Fisher's exact test was used to examine the dependency among epithelial expression programs, by detecting co-occurring or mutually exclusive pairs of expression programs. The likelihood and strength were indicated by the odds ratios and their P-values.

**Scoring of gene expression signatures in T cells**. Treg signature was derived from differentially expressed genes across all CD4+ T cell subtypes. Pearson correlation between the reference gene IL2RA and all other genes across CD4+ T cells using scaled expression values was analyzed. The top 30 genes having the highest correlation with the reference gene were defined as Treg signature genes. For CD8+ T cells, the HAVCR2 gene was chosen as the reference gene for defining the exhaustion signature using the same method, and the cytotoxicity signature was calculated by the cytotoxicity associated genes (PRF1, IFNG, GNLY, NKG7, GZMB, GZMA, GZMH, KLRK1, KLRB1, KLRD1, CTSW, CST7, CCL4, CCL3)[75,76]. We computed signature scores for individual cells using AddModuleScore function in Seurat.

**Analysis of T cell receptors**. The TCR sequences of single cell were processed using the cellranger vdj (10x Genomics, version 2.1.1) and aligned to the reference cellranger-vdj-GRCh38-alts-ensembl-2.0.0. In total, 87% of annotated T cells were assigned with a TCR sequence. Each unique TCR sequence was defined as a clonotype and T cells derived from the same cell clone were considered to share the same clonotype. Clonal cells were considered as cells with clonotype presenting in at least two cells (clone size ≥2). To demonstrate the lineage relationship between T cell subtypes, we profiled the shared TCR clonotypes between T cell subtype pairs. It was measured as the proportion of clonotypes belonging to a primary phenotype subtype (rows) shared with the secondary phenotype subtype (columns).

**Coculture experiment of tDC and CD8+ T cells**. Fresh ESCC tumor was placed in RPMI-1640 medium with 20% FBS on ice immediately after surgical resection. Tumor was rinsed with PBS, gently cut into small pieces on ice and digested in RPMI-1640 medium containing 2 mg/ml collagenase IV and 0.5 mg/ml hyaluronidase for 1 h at 37 °C. The digested cell suspension was subsequently filtered through a 70-μm cell strainer and incubated in 1X red blood cell lysis buffer on ice for 5 min. The remaining cells were suspended in 1000 μl of PBS containing 1% FBS after washing once with the same medium. Single-cell suspension was stained with CD45-FITC (BD Biosciences, 555482, dilution 1:100), CCR7-PE (BioLegend, 353203, dilution 1:100) and CD274-PerCP/Cy5.5 (BioLegend, 329737, dilution 1:100). Tumor-infiltrating tDCs (CD45+CCR7+CD274+) were sorted by using a FACSAria flow cytometer for downstream coculture experiments. Peripheral blood was obtained from the same patient, autologous CD8+ T cells were enriched by Human CD8+ T cell enrichment cocktail (Stemcell, negative isolation) and then collected by Ficoll-Paque density gradient centrifugation. Autologous CD8+ T cells (about 10⁵) were incubated in T cell expansion medium (RPMI-1640 medium with 1000 U/ml human IL-2) in 96-well plate, tDCs were co-cultured with autologous CD8+ T cells (tDC: T cell ratio of 1:10) with CD3/CD28 T cell activator (Gibco, Dynabeads Human T-Activator). For inhibition of the PD1/PD-L1 pathway, PD-L1 antibody (Bio X Cell, BE0285, dilution 100 μg/ml) was added to the well at a final concentration of 100 μg/ml. T cell number was calculated using hemocytometer at day 1 and day 2. The coculture supernatant was collected at day 2, IL-2 and IFN-γ levels were measured with cytometric bead arrays (BD Biosciences, CBA Human IFN-γ and IL-2 Flex Set).

**Analysis of correlations among the ESCC ecosystem components**. Pairwise Spearman correlation between different immune cells, immune and nonimmune stromal cells, and stromal cells and epithelial expression programs were examined to indicate the interactions among the ESCC ecosystem components. Cell types within the same major lineage develop from the same common progenitor cells and have their own differentiation trajectories. The proportion of cell subtypes was normalized within the major lineage groups including CD4+ T cells, CD8+ T cells, B cells, DCs, non-DC TIMs, fibroblasts/pericytes and endothelial cells, respectively. We tested whether the dissociation conditions would influence the cell proportion levels. According to average expression levels of 132 dissociation-induced genes[77],

we divided the samples into two groups with high or low expression level of dissociation related genes, namely high or low dissociation group. We compared the mean and coefficient of variation (CV) of cell proportions for each cell type between two groups and found little difference between two dissociation conditions. These results are shown in Supplementary Fig. 5k–m.

The correlation coefficients were calculated based on the proportion of each cell subtype to show the interplays among different immune cell subtypes or stromal cell subtypes. Spearman correlation between the epithelial program scores and the proportions of stromal cells was calculated to indicate the interactions of epithelial cells and stromal cells. Significant correlations (P < 0.05) were visualized by heatmap.

**Construction of single-cell trajectories**. Single cells assigned to fibroblasts and pericytes were used to construct the diffusion map and perform the partition-based graph abstraction (PAGA) analysis. Diffusion-map dimensionality reduction was performed using the RunDiffusion function in Seurat and the first 3 components of the diffusion map were calculated. The PAGA analysis was performed with Scanpy package (version 1.2.2)[78] to quantify the connectivity of fibroblast/pericyte subtypes.

**Analysis of the cell–cell interaction**. The cell-cell interaction between endothelial cells and fibroblasts/pericytes was predicted with CellphoneDB (version 2.0)[79]. The interaction between two cell types was measured based on the expression of the receptors in one cell type and ligands in the other cell type. Based on random permutations of cell types, the mean of the average receptor/ligand expression level in the interacting pair and P-value for the likelihood of cell-type specificity of the given interacting pair were calculated.

**Prediction of activated TFs using SCENIC analysis**. The activities of gene regulatory networks (GRNs) and TFs were identified by the SCENIC python workflow (version 0.9.1) using default parameters (https://github.com/aertslab/pySCENIC). The human TF gene list was collected from the same resource. Activated TFs were identified in the Binary matrix, and the differentially activated TFs were selected using the Wilcoxon test based on the AUC matrix (FDR < 0.05 and Fold change >2).

**Whole-exome and whole-genome sequencing and data analysis**. Genomic DNA from blood, adjacent normal tissue and tumor samples of scRNA-seq cohort was extracted using the QIAamp DNA mini Kit (Qiagen). The sequencing libraries for WGS were constructed using Tn5 transposase and sequenced on HiSeq XTen (Illumina) with 2 × 150 bp paired-end mode. WES libraries were constructed using NEBNext Ultra DNA Prep Kit for Illumina (New England Biolabs), followed by exome enrichment using SureSelect Human All Exon V6 (Agilent Technologies). The WES libraries were sequenced on NovaSeq 6000 (Illumina) with 2 × 150 bp paired-end mode. The mean sequencing depth for WES samples was about 150X (for tumor samples) while the depth was about 1X for WGS samples.

The baseqCNV pipeline[80] was used for CNV analysis of WGS results. In brief, the raw reads were aligned to reference genome (GRCh38) using BWA-MEM (version 0.7.17). The read counts in each dynamic bin were counted, normalized with bin size and sequencing depth, then corrected GC bias using LOWESS smoothing. For these bulk samples, the ploidy number was set to 2 and the DNA copy number data was segmented to identify abnormal genomic regions.

Whole-exome sequencing results of tumor and adjacent normal tissues were mapped to the GRCh38 reference genome using BWA-MEM (version 0.7.17). The WES data from 46 ESCC was retained for the subsequent analysis. Deduplication was performed with Picard (http://broadinstitute.github.io/picard/; version2.18.16) and base quality recalibration was done using BQSR module of GATK (version 4.0). The variants were called with mutect2 module of GATK. The high-quality and reliable single nucleotide variations required at least 100x coverage, minimal allele frequency (AF) ≥ 10% in tumor and AF < 2% in normal tissue. The annotation of variants was conducted with ANNOVAR (version 2017jun)[81]. Mutational signatures were extracted using the R package maftools and the previously published data were combined to increase the power and accuracy[5]. The frequency of each signature was estimated in 46 ESCC samples. We identified 4 mutational signatures showing high similarities with COSMIC signatures (all cosine similarity >0.8), including alcohol drinking related signature A (Sig. A), tobacco smoking related signature B (Sig. B), ageing related signature C (Sig. C) and APOBEC related signature D (Sig. D).

**Analysis of effects of genomic alterations on ESCC ecosystem components**. To examine whether the somatic driver gene mutations identified by WES have impacts on the epithelial expression programs and components in the ESCC TME, the Wilcoxon test was performed to compare the epithelial program scores and TME subtype proportions between ESCC samples with or without the mutations.

**RNA sequencing in bulk tumor tissues**. Bulk RNA-seq was conducted in samples obtained from the Discovery cohort of 139 ESCC patients for survival analysis. The

library preparation and sequencing were performed as previously described[5]. Briefly, RNA from the tissue samples was extracted using the Allprep RNA Kit (Qiagen) and sequenced by Illumina NovaSeq 6000 with a total of 10 Gb data. RNA-seq data were mapped to GRCh38 reference genome and the gene expression values were calculated using RSEM[82].

**Immunohistochemical staining of marker genes**. Tissue microarrays were prepared from 226 ESCC tumors in Validation cohort 2. Slides were incubated with antibody against AGR2 (Abcam, ab76473, dilution 1:1000), CXCL17 (Proteintech, 18108-1-AP, dilution 1:50) or MUC20 (Abgent, AP7830b, dilution 1:100) and detected with the Dako REALTM EnVisionTM Detection System (K5007). DAB (3,3′-diaminobenzidine) staining intensity was analyzed by color deconvolution algorithms in inForm software (PerkinElmer; version 2.4.2). The H-score method was used to evaluate the percentage of positive stained cells and staining intensity. The average intensity of staining in positive cells was assigned as an intensity score (0, none; 1, weak; 2, moderate, and 3, strong). The score was obtained by the formula: 3 × % of strongly stained cells + 2 × % of moderately stained cells + 1 × % of weakly stained cells.

**Analysis of the correlation between gene expression levels and ESCC survival time**. The correlations between the expression levels of genes in the epithelial expression programs and ESCC survival time were examined in the Discovery cohort ($N = 139$) and validated in two Validation cohorts ($N = 94$ and $N = 226$, respectively). For discovery, the RSEM normalized read counts of bulk RNA-seq were log2 transformed and the possible effects of TME compositions were removed using the R package estimate (version 1.0.13)[83]. The expression levels of each gene were normalized by tumor purity and the median level of each gene was used as dividing point of high (≥median) or low (<median) expression for survival analysis. The median survival time in patients with high or low expression of the genes was estimated by Kaplan–Meier method and the significance of difference was examined by log-rank test. We identified 14 genes significantly correlated with ESCC survival (all $P < 0.05$) in the Discovery cohort. The significantly correlated gene expressions were further validated using the same method in Validation cohort 1, which was described in our previous study[5] and in Validation cohort 2 using H-scores of the protein staining in tissue microarray of ESCC samples. The minimum log-rank P-value analysis was performed to demonstrate potential H-score threshold values of the interest proteins for prognostic stratification. The threshold with the smallest log-rank P-value is selected.

**Reporting summary**. Further information on research design is available in the Nature Research Reporting Summary linked to this article.

## Data availability

The raw RNA and DNA sequencing data of this study have been deposited into the Gene Expression Omnibus (GEO) with accession number GSE160269 and Sequence Read Archive with accession number SRP327447, respectively. The raw sequencing data are also available from the Genome Sequence Archive of Beijing Institute of Genomics, Chinese Academy of Sciences with accession number HRA000195. Gene expression matrix of ESCC and paired adjacent normal samples are also available from the GEO with accession number GSE160269. VCF files containing variants called from ESCC genomes have been deposited into the European Variation Archive with accession ID PRJEB41091. We obtained the hallmark gene sets from the MSigDB database (http://www.gsea-msigdb.org/gsea/msigdb/genesets.jsp?collection=H). The remaining data are available within the article, supplementary information and Source data. Source data are provided with this paper.

## Code availability

Example scripts to process and analyze data are available at https://github.com/friedpine/scRNASeq_ESCC. Detailed information will be available from the corresponding author upon reasonable request.

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

## Acknowledgements

This project was supported by the National Key Research and Development Program of China. (2016YFC1302700 to C. Wu), National Natural Science Foundation of China (81725015 to C. Wu, 81988101 to D.L. and C. Wu, 21675098 to J.W.), Beijing Out-standing Young Scientist Program (BJJWZYJH01201910023027 to C. Wu), Medical and Health Technology Innovation Project of Chinese Academy of Medical Sciences (2019-I2M-2-001 to D.L. and C. Wu), Ministry of Science and Technology of China (2018YFA0800200 to J.W.), 2018 Beijing Brain Initiation (Z181100001518004 to J.W.).

## Author contributions

D.L., C. Wu, and J.W. conceptualized and supervised this study. X. Zhang, L.P., Y. Luo, S.Z., Y.P., Y.C., Q.C., W.F., and J.Y. contributed to the study design and performed most experiments. W.G., M.S., Y.X., X.N., C. Wang, Y. Liu., and W.T. were engaged in patient recruitment and sample acquisition. J.Y., Q.C., W.F., Y.C., X. Zhao, L.C., C.Z., and X.Y. collected single cells and generated sequencing data. X. Zhang, L.P., Y. Luo, Y.M., and Y.W. contributed to bioinformatics and statistical analysis. L.P., Y. Luo, and Y.S. per-formed the biological analysis and interpretation. X. Zhang, L.P., and Y. Luo drafted the manuscript. D.L., C. Wu and J.W. reviewed and prepared the final manuscript. All authors approved the final manuscript.

## Competing interests

The authors declare no competing interests.
