## [Peer Review File · Nature Communications]

Dissecting esophageal squamous-cell carcinoma ecosystem by single-cell transcriptomic analysisEditorial Note: In the second round of review, Reviewer #2 was unavailable to comment on the revised manuscript; therefore we recruited Reviewer #4 and #5 to comment on the author's response to Reviewer #2.

REVIEWER COMMENTS

Reviewer #1 (Remarks to the Author): Expert in immunogenomics

In this study, the authors took the single-cell transcriptome (scRNA-seq) and TCR (scTCR-seq) profiling approach, together with bulk WES and WGS data, to understand epithelial cells of a variety of expression program and the tumor microenvironment (TME) among 60 patients with esophageal squamous cell carcinoma (ESCC), as first of its kind. The epithelial program characterized by intratumoral and intertumoral heterogeneity among ESCC patients that cannot be otherwise dissected from conventional bulk RNA sequencing. With a rich dataset of cell types in the TME with detailed annotation, the study also provided reasonable interactions among the TME and epithelial cells. Such interactions would lead to varying degrees of immune infiltration and cancer progression, which had not been reported before in ESCC. More importantly, the authors identified three genes derived from epithelial programs and TME characters that would possibly affect patient survival. Overall, the work provided solid and valuable findings for ESCC research. Several issues need to be addressed as listed below:

Major points:

- 1) In Fig2, the authors annotated the epithelial cells into Group 1 and Group 2 clusters based on the distribution of the cluster among patients. Group 1 cluster had higher inter-tumor heterogeneity levels than Group 2 cluster. This is an interesting analysis to see the intertumoral heterogeneity. When considering cell types in the TME, were there any differences in cell type composition among patients with more or less intratumor heterogeneity?
- 2) The authors generated a nice scRNA-seq data resource and defined multiple cell subtypes in the TME. But how are TME cell type distribution or changes correlated with patients' clinical characteristics?
- 3) The authors explored the associations between driver gene mutations like TP53 and NOTCH1 and the tumor ecosystem. Were the mutational signatures based on WES data associated with any TME cell type distributions?
- 4) The authors identified that three genes in mucosal epithelial program were associated with patients' survival. It will be important to note how the expression of these three genes were effected by patients' stage or TME cell type distributions?

Minor concerns:

- 1) All the statistical tests and details should be explicitly noted in the method section.
- 2) The authors should deposit the raw data as well as the processed single-cell data as an expression matrix (cells by genes) into the public repository.

Reviewer #2 (Remarks to the Author): Expert in single-cell RNA-seq

This is a comprehensive, interesting and well put together manuscript by Zhang et al where they use single cell transcriptomics together with genomic sequencing to dissect the complexity and heterogeneity of Esophageal squamous-cell carcinoma (ESCC). The authors identify distinct expression programs in the ESCC malignant epithelial cells and establish a primary association framework that allows them to correlate mutational signatures/genomic alterations with that of the gene expression programs in both malignant cells as well as the immune/non-immune stroma in the tumor microenvironment. Overall the manuscript is well written, the analyses are unique and robust, and the conclusions are generally supported by the data. However, the major concern that I have is that a vast majority of the conclusions are based on descriptive and correlative data that lack any functional validation. The manuscript could also greatly benefit from having spatial transcriptomics or immunofluorescence-based clinical validation of the markers/gene expression programs identified in the study. Such validation could be very revealing in terms of understanding the suggested cell-cell interactions and how they change in the context of space and time (tumor staging).

Specific comments:

Fig 2 and Supp Fig 2:

- The two distinct pathway activities between Group 1 and Group 2 clusters are particularly interesting in light of the fact that Wnt/bcat signaling (and myc which is also downstream of Wnt signaling) has been associated with immune evasion/exclusion phenotypes. Is that what defines the Group1 v/s 2 Group 2 clusters? If the authors were to stratify the tumor epithelial cells based on Wnt activity, would they separate into the same two groups?
- Are the mutational signatures distinct between group 1 and group 2 clusters?

Fig. 4 and Supp Fig. 4:

- The identification of the tDC and their correlation with an immunosuppressive signature is really important. Can the authors functionally test this phenomenon in co-culture assays to demonstrate that the tDCs expressing the checkpoint ligands indeed block the activation and proliferation of cytotoxic T cells.
- The authors should also use CellPhoneDB to identify the ligand receptor interactions between tDC and CD8/CD4 T cells and perhaps also with the Tregs? Are these interactions absent or different for cDCs and TAMs? It would be interesting to see whether these tDCs are also involved in the recruitment of Tregs to generate an immunotolerant microenvironment.

Fig. 5:

- Some of the CAF clustering appears to be over segmented and may not represent true clusters/cell states. It is difficult to see how CAF3 is different from CAF4 based on the clustering in Fig 5a, or the dot plot looking at the top DEGs in the CAF clusters in 5b. the authors should provide a better rationale to distinguish between the CAF3 and CAF4.
- Based on the DEGs, can the authors identify gene regulatory networks that may be driving the transition from pericytes to CAF2 to CAF3/4 states? The authors can try to cluster the cells based on regulon activity (SCENIC) and identify potential mechanisms involved in determining these trajectories. Correlating the GRN with the specific interactions that they see with TECs could potentially uncover the molecular mechanisms that dictate the CAF identities.
- Fig. 5f: Again, the endothelial clusters look very diffuse and oversegmented. Authors may want to decrease the resolution parameters and re-cluster the ECs. It is not clear as to why the NEC2 cluster is coming from the normal samples?

Finally, I would strongly encourage the authors to validate some of the tumor epithelial, EC, tDC, CAF and CD4/8 T/NKT/Treg cells markers (activation, exhaustion) using RNA-Fish (spatial transcriptomics or RNAscope) and immunofluorescence on FFPE or frozen slides. Demonstrating the spatial configuration of the proposed cell-cell interactions in different stages (I versus III/IV) would not only serve as a strong validation of the conclusions, but also illuminate mechanistic insights into the progression of ESCC.

Reviewer #3 (Remarks to the Author): Expert in esophageal cancer genomics

Xiannian Zhang et al. performed a comprehensive analysis of ESCC, including scRNA-seq and genotyping. They deciphered not only tumor cells, but also immune and stromal cells at a single-cell level. The topic is interesting and worthy of investigation. However, the manuscript almost feels too comprehensive, lacking a strong focus.

> What are the major claims of the paper?

They identified 8 common expression programs from cancer cells and discovered 43 cell types, including immune and stromal cell subtypes, and linked cancer genotypes with the composition of such cell types. They also found expression markers related to survival time in patients.

> Are the claims novel? If not, please identify the major papers that compromise novelty

The characterization of TILs, TIMs, and CAFs is based on previous knowledge, and there are few novel findings. The characterization of tumor cells and their link to genotypes are novel, and they should focus more on this part.

> Will the paper be of interest to others in the field?

In part, yes, but some pieces of the paper feel obsolete.

> Will the paper influence thinking in the field?

The paper provides facts that link tumor genotypes to their microenvironment at a single-cell level. However, some analyses are not convincing.

> Are the claims convincing? If not, what further evidence is needed?

> Are there other experiments that would strengthen the paper further? How much would they improve it, and how difficult are they likely to be?

In the clustering analysis of cancer cells, they first clustered all cancer cells derived from all subjects and got 30 subtypes. However, in the later sections, they clustered cancer cells by a different way and extracted just 8 subtypes. I understand that they want to claim the presence of patient-specific cancer cell subtypes in the former analysis, but this seems not important and even confusing. From Line 364, they performed WES on ESCC that were also analyzed by scRNA-seq. Considering nearly all ESCCs have TP53 mutations, the frequency of TP53 mutated cancer seems lower than

expected. According to Suppl. Fig. 6a, some TP53-wild cancers have no other drivers and few somatic mutations. There is a possibility of false negative due to low sensitivity of WES. Because genotyping of cancer affects the result of following analyses, targeted-captured deep sequencing of driver genes is recommended.

> Are the claims appropriately discussed in the context of previous literature?

Yes, their claims are in the context of previous studies in general.

> If the manuscript is unacceptable in its present form, does the study seem sufficiently promising that the authors should be encouraged to consider a resubmission in the future?

Yes. I think the authors should focus more on their novel findings. Some additional experiments are also recommended.

> Should the authors be asked to provide further data or methodological information to help others replicate their work? (Such data might include source code for modelling studies, detailed protocols or mathematical derivations).

They should make their scRNA-seq and genotyping data publicly available to help others replicate their work.

Responses to the Reviewers' Comments

Reviewer #1 (Remarks to the Author): Expert in immunogenomics

In this study, the authors took the single-cell transcriptome (scRNA-seq) and TCR (scTCR-seq) profiling approach, together with bulk WES and WGS data, to understand epithelial cells of a variety of expression program and the tumor microenvironment (TME) among 60 patients with esophageal squamous cell carcinoma (ESCC), as first of its kind. The epithelial program characterized by intratumoral and intertumoral heterogeneity among ESCC patients that cannot be otherwise dissected from conventional bulk RNA sequencing. With a rich dataset of cell types in the TME with detailed annotation, the study also provided reasonable interactions among the TME and epithelial cells. Such interactions would lead to varying degrees of immune infiltration and cancer progression, which had not been reported before in ESCC. More importantly, the authors identified three genes derived from epithelial programs and TME characters that would possibly affect patient survival. Overall, the work provided solid and valuable findings for ESCC research. Several issues need to be addressed as listed below.

Response: We thank the Reviewer for his/her very positive comments on our study. We are happy to address the issues the Reviewer raised.

Major points:

1) In Fig2, the authors annotated the epithelial cells into Group 1 and Group 2 clusters based on the distribution of the cluster among patients. Group 1 cluster had higher inter-tumor heterogeneity levels than Group 2 cluster. This is an interesting analysis to see the intertumoral heterogeneity. When considering cell types in the TME, were there any differences in cell type composition among patients with more or less intratumor heterogeneity?

Response: Thanks for the thoughtful question. We found that heterogeneity level is associated with patients' age and drinking status, which has been described (page 6, lines 132-136 and Supplementary Figure 2c). Per the comment, we have looked at the relationships between the intratumoral heterogeneity levels and their TME status and found that the proportions of different cell types in TME did not significantly differed between Group 1 and Group 2 clusters as shown in the figure below.

2) The authors generated a nice scRNA-seq data resource and defined multiple cell subtypes in the TME. But how are TME cell type distribution or changes correlated with patients' clinical characteristics?

Response: We associated the proportions of each cell type in TME with various clinical factors including age, sex, smoking status and drinking status (see Figure below). We found that the immunity status of T cells and proportions of non-immune stromal cells were associated with clinical characteristics. For example, male patients who smoked had higher cytotoxicity and exhaustion scores but fewer CAF1. Male patients had more type 2 tumor endothelial cells (TEC2) but fewer type 2 normal endothelial cells (NEC2). Furthermore, we found that patients who were alcohol drinker had higher cytotoxicity score of T cells compared with patients who were nondrinkers. We have added these results in revised manuscript (page 16, lines 368-370; Supplementary Figure 5o). We hope these revisions are satisfactory.

3) The authors explored the associations between driver gene mutations like TP53 and NOTCH1 and the tumor ecosystem. Were the mutational signatures based on WES data associated with any TME cell type distributions?

Response: Per the comment, we have looked at the associations between 4 mutational signatures based on WES data and TME subtypes. We found that the 4 mutational signatures are highly similar to COSMIC signatures (all cosine similarity > 0.8) of alcohol drinking related signature A (Sig. A), tobacco smoking related signature B (Sig. B), aging related signature C (Sig. C) and APOBEC related signature D (Sig. D). We found that ESCC tumors with higher Sig. A level had less infiltration of T_{EFF} but more NEC1. ESCC tumors with higher Sig. B level had increased TEC3 and Treg scores. ESCC tumors with higher Sig. C had increased NK/NKT but decreased CAF3 while tumors with higher Sig. D had more activated B cells and TAM but decreased monocytes. We have added these results to revised manuscript (page 17, lines 399-401; Supplementary Figure 6f). For your review convenience, the data is also shown below.

4) The authors identified that three genes in mucosal epithelial program were associated with patients' survival. It will be important to note how the expression of these three genes were affected by patients' stage or TME cell type distributions?

Response: This comment is somewhat confusing for us. The expression levels of the three genes are not significantly different among various disease stages or affected by the distribution of TME cell types.

Minor concerns:

1) All the statistical tests and details should be explicitly noted in the method section.

Response: We have added statistical tests and details in revised manuscript.

2) The authors should deposit the raw data as well as the processed single-cell data as an expression matrix (cells by genes) into the public repository.

Response: We have deposited the raw data and expression matrix into the public repository (GSA: HRA000195 and GEO: GSE160269). This information has been provided in revised manuscript (page 35, line 841-848).

Reviewer #2 (Remarks to the Author): Expert in single-cell RNA-seq

This is a comprehensive, interesting and well put together manuscript by Zhang et al where they use single cell transcriptomics together with genomic sequencing to dissect the complexity and heterogeneity of Esophageal squamous-cell carcinoma (ESCC). The authors identify distinct expression programs in the ESCC malignant epithelial cells and establish a primary association framework that allows them to correlate mutational signatures/genomic alterations with that of the gene expression programs in both malignant cells as well as the immune/non-immune stroma in the tumor microenvironment. Overall the manuscript is well written, the analyses are unique and robust, and the conclusions are generally supported by the data. However, the major concern that I have is that a vast majority of the conclusions are based on descriptive and correlative data that lack any functional validation. The manuscript could also greatly benefit from having spatial transcriptomics or immunofluorescence-based clinical validation of the markers/gene expression programs identified in the study. Such validation could be very revealing in terms of understanding the suggested cell-cell interactions and how they change in the context of space and time (tumor staging).

Response: Thanks for the positive general comment. Per the suggestion, we have conducted cell-cell interaction analysis for tDC and other myeloid subtypes and performed co-culture experiment of tDC and CD8+ T cells which validated the immunosuppressive role of tDC through expression of PD-L1. We have also re-annotated the endothelial cells and explored the GRNs of CAFs. We agree that the manuscript could greatly benefit from having spatial transcriptomics; however, because the present manuscript already contains huge dataset and many figures, we would like to have spatial transcriptomics data in the next report.

Specific comments:

Fig 2 and Supp Fig 2:

- The two distinct pathway activities between Group 1 and Group 2 clusters are particularly interesting in light of the fact that Wnt/bcat signaling (and myc which is also downstream of Wnt signaling) has been associated with immune evasion/exclusion phenotypes. Is that what defines the Group1 v/s 2 Group 2 clusters? If the authors were to stratify the tumor epithelial cells based on Wnt activity, would they separate into the same two groups?

Response: Per the suggestion, we have examined if Wnt activity can separated the same two groups. The WNT activity (score) was defined on the basis of the average expression level of 42 genes in the Wnt/bcat pathway from GSEA database for epithelial cells within each sample. We found that the WNT scores are correlated with relative heterogeneity level ($R = 0.33, P = 0.018$; Figure A below), suggesting that it may stratify tumor cells in terms of heterogeneity levels, especially in samples having high WNT activity. For example, we found that among 6 samples having the highest WNT score, 5 are classified into Group 1 (Figure B below). However, it is not the case in samples having low WNT score and low heterogeneity level. These results suggest that not only the Wnt/bcat pathway, but also other pathways might be involved in making heterogeneity of ESCC cells.

- Are the mutational signatures distinct between group 1 and group 2 clusters?

Response: We have looked at the proportions of mutational signatures in two group samples and found no significant differences except for Sig. A (related to drinking), which is marginally significant ($P = 0.09$) as shown in the following figure. This result is consistent with the result showing that drinker patients had higher

heterogeneity level than non-drinker patients (Supplementary Figure 2c). We have added this result in revised manuscript (page 17, line 399-401; Supplementary Figure 6e).

Fig. 4 and Supp Fig. 4:

- The identification of the tDC and their correlation with an immunosuppressive signature is really important. Can the authors functionally test this phenomenon in co-culture assays to demonstrate that the tDCs expressing the checkpoint ligands indeed block the activation and proliferation of cytotoxic T cells.

Response: Per the suggestion, we have performed co-culture assays using tDCs isolated by FACS from ESCC tissue of one patient on the basis of the makers of CD45+CCR7+CD274+ (CD45+-FITC, CCR7+-PE, CD274+-PercP-Cy5.5) as shown in the following Figure A. The isolated tDCs were then co-cultured with autologous CD8+ T cells isolated from human peripheral blood sample with stimulation by anti-CD3/28 at the ratio of tDCs:T Cells-1:10. Our scRNA-seq results show that FACS sorting with the three markers generated cells contains >85% tDCs. Co-cultured assays have shown that tDCs significantly inhibited the proliferation of cytotoxic CD8+ T cells stimulated with anti-CD3/CD28 (Figure B). Furthermore, CD8+ T cells co-cultured with tDCs had diminished production of IL-2 and IFN- γ (Figure C). We have also examined the potential mechanism of tDCs in inhibiting T cells and found that when anti-PD-L1 antibody (Durvalumab) was presented in the co-culture, the reduced proliferation of anti-CD3/28 stimulated CD8+ T cells and suppressed production of IL-2/IFN- γ could be rescued (Figures B and C), suggesting that the suppressing effect of tDC on CD8+ T cell proliferation and activation is likely mediated through PD1 and PD-L1 interaction.

Our results described above are in line with previous findings. For example, it has been reported that tDCs in patents with systemic lupus erythematosus repress autologous T cell activation (Obreque et al., 2017); the immune checkpoint ligands we identified in tDCs in the microenvironment of ESCC have been reported as markers in tDC (Morelli & Thomson, 2007). We have added these additional results in the revised manuscript (page 11, lines 261-268; Figures. 4i and j) and a small paragraph to the Methods part to describe the detail of the

co-culture assays (page 29, lines 698-717).

• The authors should also use CellPhoneDB to identify the ligand receptor interactions between tDC and CD8/CD4 T cells and perhaps also with the Tregs? Are these interactions absent or different for cDCs and TAMs? It would be interesting to see whether these tDCs are also involved in the recruitment of Tregs to generate an immunotolerant microenvironment.

Response: We have conducted CellPhoneDB analysis to identify the ligand-receptor interactions between T cell subtypes including Treg and the 4 types of immune-regulatory myeloid cells i.e., tDC, cDC, pDC and TAM (see Figure below). tDC showed stronger interactions with multiple T cell subtypes than cDC, pDC and TAM. The suppressive interactions included ligand-receptor pairs such as PVR-TIGIT, PD1-PD-L1/2, CTLA4-CD86 and BTLA-TNFRSF14. Collectively, tDCs present more immunosuppressive effects on multiple T cell subtypes compared with other DC cells and TAM.

We have also identified two well-known ligand-receptor pairs involving in migration and recruitment of Treg: CCL17-CCR4 and CCL22-CCR4. Because these two interactions were most significant in tDCs, suggesting that tDCs may also involve in Treg recruitment. We have added these results in revised manuscript (page 11, lines 259-261; Supplementary Figure 4i).

• Some of the CAF clustering appears to be over segmented and may not represent true clusters/cell states. It is difficult to see how CAF3 is different from CAF4 based on the clustering in Fig 5a, or the dot plot looking at the top DEGs in the CAF clusters in 5b. the authors should provide a better rationale to distinguish between the CAF3 and CAF4.

Response: To determine an appropriate clustering parameter, we adjusted the granularity of clustering analysis by using a series of resolution values ranging from 0.3 to 0.6 for the FindClusters function in Seurat package as shown in the following Figure A. The existence of all clusters including CAF3 and CAF4 are consistent with rougher resolutions (Res = 0.6, 0.5 and 0.4). Under a very coarse resolution (Res = 0.3), the CAF3 cannot be differentiated from CAF4 and they merge into a single cluster. However, we found that clustering analysis under such resolution may overlook some real biological distinction. For example, VSMC is a distinctive cell type with remarkable expression of marker genes *MYH11* and *CNN1* (Figure 5b in our original manuscript); however, with this coarse resolution, VSMC and pericytes are classified into a same single cell type (Res = 0.3), indicating that this cell type annotation is not suitable. Thus, we used a moderate clustering resolution (Res = 0.5), which effectively differentiates the real biological subtypes such as VSMC and CAF3.

We found that CAF3 may be the transition or precursor subtype of CAF4. Monocle Analysis of CAF1/2/3 showed that CAF3 indeed mediates the transformation of CAF1 to CAF4 as shown in the following Figure B. Also, the marker genes of CAF1 and CAF4 (e.g., *CCL2*, *CXCL1*, *ACTA2* and *POSTN*) had intermediate expression levels in the CAF3 cluster (Figure C). Some subtle but evident differences between the CAF3 and CAF4 clusters can be found. The DEG analysis using Wilcoxon test showed 71 marker genes for CAF3 and 11 for CAF4 (FDR < 0.05, Fold change > 1.5; Figure D). CAF3 expressed high levels of cytokines such as *CXCL1/2* and *CCL2* and extracellular matrix proteinases such as *MMP1* and *MMP3*. Collectively, these results indicate that the CAF3 cluster we identified is really presented but not resulted from inappropriate clustering analysis.

• Based on the DEGs, can the authors identify gene regulatory networks that may be driving the transition from pericytes to CAF2 to CAF3/4 states? The authors can try to cluster the cells based on regulon activity (SCENIC) and identify potential mechanisms involved in determining these trajectories. Correlating the GRN with the specific interactions that they see with TECs could potentially uncover the molecular mechanisms that dictate the CAF identities.

Response: We appreciate this thoughtful suggestion. We have performed additional analysis and detected 21 genes whose expression levels were higher in CAF2 compared with CAF3/4. Gene ontology analysis revealed the enrichment of collagen degradation, JAK-STAT and MAPK cascade pathways (please refer the following figure).

We have further performed the SCENIC analysis on pericyte, CAF2 and CAF3/4 (see Figure below). The GRN specifically activated in each cell type are filtered (FDR < 0.05, Fold change > 2). As a result, 18 GRN (represented by the name of core transcription factors) have been detected for CAF2. We have also found several ETS family transcription factors (e.g., EHF, ELF3 and ETS2) that are significantly activated which are the downstream effectors of MAPK pathway. Considering the PDGFB-PDGFRB interaction may trigger MAPK signaling pathway, the results of SCENIC analysis may indicate candidate targets of the interaction between TECs and pericyte-to-CAF2. We have added these results in revised manuscript (page 15, lines 348-350; page 31, lines 749-754; Supplementary Fig. 5j).

• Fig. 5f: Again, the endothelial clusters look very diffuse and oversegmented. Authors may want to decrease the resolution parameters and re-cluster the ECs. It is not clear as to why the NEC2 cluster is coming from the normal samples?

Response: Per the suggestion, we have carefully reanalyzed the endothelial clusters by using a series of resolution values from 0.5 to 0.2. Indeed, the clustering is significantly changed by the resolution parameters (please refer the following figures). We found that although resolution value 0.4, which was used in our original manuscript, generated possibly over-segmented clusters, a much rougher resolution value, 0.2, results in clusters that may obscure the patterns of the key NEC and TEC markers (e. g., *CCL2*, *IL6*, *ANGPT2* and *ESM1*). Finally, we have found that resolution value 0.3 is the best parameter, which identified 6 clusters of endothelial cells in the ESCC TME (i.e., NEC1–3 and TEC1–3) that fit well the expression patterns of the key markers. Therefore, in revised manuscript, we have changed the statements about the endothelial clustering results (page 14, lines 329-332; Figure 5f; Supplementary Fig. 5h) by adopting resolution value 0.3.

We are confused by the Reviewer’s question ‘It is not clear as to why the NEC2 cluster is coming from the normal samples.’ NEC2 refers normal endothelial cell cluster 2 that should be coming from the normal samples.

Finally, I would strongly encourage the authors to validate some of the tumor epithelial, EC, tDC, CAF and CD4/8 T/NKT/Treg cells markers (activation, exhaustion) using RNA-Fish (spatial transcriptomics or RNAscope) and immunofluorescence on FFPE or frozen slides. Demonstrating the spatial configuration of the proposed cell-cell interactions in different stages (1 versus III/IV) would not only serve as a strong validation of the conclusions, but also illuminate mechanistic insights into the progression of ESCC.

Response: We agree with the suggestions and are aware of the importance of the validation. However, due to already having huge dataset in the present manuscript, which focuses on deciphering for the first time the compositions in the ESCC tumor microenvironment, we would like to ask the Reviewer and Editor to allow us to address this issue in the next studies using a spatial transcriptome technique. The further study is on the way.

Reviewer #3 (Remarks to the Author): Expert in esophageal cancer genomics

Xiannian Zhang et al. performed a comprehensive analysis of ESCC, including scRNA-seq and genotyping. They deciphered not only tumor cells, but also immune and stromal cells at a single-cell level. The topic is interesting and worthy of investigation. However, the manuscript almost feels too comprehensive, lacking a strong focus.

Response: We appreciate the comment. Because this is the first scRNA-seq analysis of ESCC, we aimed to provide a comprehensive landscape of ESCCs and their tumor microenvironments (TME). However, except for the comprehensive description, we have also made efforts to establish the associations between somatic mutations and changes in TME or transcriptomic changes at single cell level and patient clinical characteristics. In revised manuscript, we have also added some experimental validation data to address the potential mechanisms underlying the immunosuppressive ESCC TME.

> What are the major claims of the paper?

They identified 8 common expression programs from cancer cells and discovered 43 cell types, including immune and stromal cell subtypes, and linked cancer genotypes with the composition of such cell types. They also found expression markers related to survival time in patients.

Response: Thanks for this brief summary of our study findings.

> Are the claims novel? If not, please identify the major papers that compromise novelty

The characterization of TILs, TIMs, and CAFs is based on previous knowledge, and there are few novel findings. The characterization of tumor cells and their link to genotypes are novel, and they should focus more on this part.

Response: Thanks for this pertinent comment. Although the characterization of TILs, TIMs and CAFs is based on previous knowledge, we have reconstructed differentiation lineages of CAF and establish their interplays with ESCC cells and other cell types in the tumor microenvironment (TME). Because, to the best of our knowledge, this is the first scRNA-seq paper on ESCC, we would like to show the landscape or full profile of ESCC and its TME.

> Will the paper be of interest to others in the field?

In part, yes, but some pieces of the paper feel obsolete.

Response: As we have stated in the above Response, this is the first paper on scRNA-seq analysis of ESCC and we therefore want to retain full data as a data source for readers who may be interested in this field.

> Will the paper influence thinking in the field?

The paper provides facts that link tumor genotypes to their microenvironment at a single-cell level. However, some analyses are not convincing.

Response: The Reviewer does not specify what analyses are not convincing. Per the comments from other Reviewers, we have performed additional experiments and analysis and reworded some parts of the results in revised manuscript. We hope that the revisions we have made have also satisfied this comment.

> Are the claims convincing? If not, what further evidence is needed?

> Are there other experiments that would strengthen the paper further? How much would they improve it, and how difficult are they likely to be?

In the clustering analysis of cancer cells, they first clustered all cancer cells derived from all subjects and got 30 subtypes. However, in the later sections, they clustered cancer cells by a different way and extracted just 8 subtypes. I understand that they want to claim the presence of patient-specific cancer cell subtypes in the former analysis, but this seems not important and even confusing.

Response: Thanks for posing the potential issue. Yes, we clustered the ESCC cells in two different ways. The first way was to cluster all cancer cells derived from all subjects and identify the inter-individual heterogeneity of the cancer that was unable to be depicted by the traditional bulk transcriptomic analysis; the second way was to elucidate the effects of the common expression programs in cancer cells on the compositions in tumor microenvironment which might cause the inter-individual heterogeneity of tumor biology and clinical outcomes in patients. So, we believe that this two-way clustering is reasonable. We hope this explanation is satisfactory.

From Line 364, they performed WES on ESCC that were also analyzed by scRNA-seq. Considering nearly all ESCCs have TP53 mutations, the frequency of TP53 mutated cancer seems lower than expected. According to Suppl. Fig. 6a, some TP53-wild cancers have no other drivers and few somatic mutations. There is a possibility of false negative due to low sensitivity of WES. Because genotyping of cancer affects the result of following analyses, targeted-captured deep sequencing of driver genes is recommended.

Response: In the present study, the mean sequencing depth of WES is about 150X (100X for normal samples and 300X for tumor samples). We detected 89.1% (41/46) of samples having *TP53* mutations and only 3 samples had no putative driver gene mutation, which are comparable to those reported in the previous studies with larger sample sizes (Chang et al., 2015; Cancer Genome Atlas Research 2017; Li et al., 2018). For example, in Chang et al. study (n=704), the frequency of *TP53* mutated ESCC samples is 85.0%, similar to the finding in the present study (89.1%). These results indicate that our WES data are reliable. Per the comment, we have performed additional analysis of mutations by enriching the potential driver genes such as *TP53* using target-captured deep sequencing, in which the median sequencing depth for *TP53* increases to >1000X. As a result, we found that the rate of called *TP53* mutations is identical to that identified by WES as shown in Figure A below. We have also analyzed the 5 samples (P12, P16, P30, P48 and P104) by visualizing sequencing data using IGV and the results

show that they indeed had no evident somatic *TP53* mutations (Figure B). Together, these results strongly indicate that the frequency of *TP53* mutations in our analysis is reliable.

> Are the claims appropriately discussed in the context of previous literature?

Yes, their claims are in the context of previous studies in general.

Response: Thanks.

> If the manuscript is unacceptable in its present form, does the study seem sufficiently promising that the authors should be encouraged to consider a resubmission in the future?

Yes. I think the authors should focus more on their novel findings. Some additional experiments are also recommended.

Response: Again, this is the first paper on scRNA-seq analysis of ESCC and therefore we want to retain all valid data as a data source for the readers who may be interested in this field. We have performed some additional experiments, for example, to validate the crosstalk between tDC and CD8+ T cells through PD1 and PD-L1 interaction, as suggested by Reviewer #2. Furthermore, we have also carried out in silico cellular ligand-receptor interaction analyses to look at the inhibitory effects of 4 immuno-regulatory myeloid cells (tDC, pDC, cDC and TAM) on T cells and found that tDC had the most inhibitory effect on multiple subtypes of T cells. These new results have been added to revised manuscript (page 11, lines 259-268; Figures. 4i and j). We hope these revisions are satisfactory.

> Should the authors be asked to provide further data or methodological information to help others replicate their work? (Such data might include source code for modelling studies, detailed protocols or mathematical derivations).

They should make their scRNA-seq and genotyping data publicly available to help others replicate their work.

Response: Yes, we have deposited the scRNA-seq and genotyping data to the public repository and the related information has been noted in revised manuscript (page 35, lines 841-848).

References

- Morelli, A. E., & Thomson, A. W. (2007). Tolerogenic dendritic cells and the quest for transplant tolerance. *Nature Reviews Immunology*, 7(8), 610-621. doi:10.1038/nri2132
- Obreque, J., Vega, F., Torres, A., Cuitino, L., Mackern-Oberti, J. P., Viviani, P., . . . Llanos, C. (2017). Autologous tolerogenic dendritic cells derived from monocytes of systemic lupus erythematosus patients and healthy donors show a stable and immunosuppressive phenotype. *Immunology*, 152(4), 648-659. doi:10.1111/imm.12806

REVIEWER COMMENTS

Reviewer #1 (Remarks to the Author):

Authors have addressed all my concerns.

Reviewer #3 (Remarks to the Author):

The authors have adequately performed targeted deep sequencing of ESCC samples to address my initial concerns.

Reviewer #4 (Remarks to the Author): Expert in single-cell RNA-seq and computational genomics

Zhang et al present the data gathered from single-cell RNA sequencing of multiple esophageal squamous cell carcinoma samples, and a few normal controls.

While I am impressed by the extent of the created dataset, I am sad to say that I find the paper virtually unreadable. This is not primarily because of the abundant grammar and language issues, although they do not help. Rather, the presented analyses lack any kind of coherent hypothesis or structure. It forces me to conclude that this manuscript mainly constitutes a descriptive report of everything the authors could think of doing with the data they gathered. If this is the case, they should consider submitting this manuscript to something like "Scientific Data" or a similar journal.

If, however, their aim is to present a biological finding, then this needs to be rewritten from the ground up, and possibly broken into multiple papers. This is also because each individual sub-analysis is done superficially. I frankly refuse to go through each of the 40 pages of this paper and list every single bioinformatics issue, but just to name a few: they fail to even describe how unsupervised clustering was performed. There is no discussion on an evaluation of robustness of clustering. The described approach for evaluating inter- and intra-tumour heterogeneity seems highly dubious - there is an obvious confounding effect when using a 2d-projection like PCA to evaluate within- and between-patient distance at the same time. Their classification of "modules" of expression seems dubious to me. The classification of epithelial cells into "Group 1" and "Group 2" seems arbitrary. There is no discussion on the effect of choosing other arbitrary thresholds of separation.

This is by no means an exhaustive list of my grievances with this paper. More importantly, though, it is futile to try and "fix" all these issues until it's clear what the point is that the authors are really trying to make. They should focus on that, do the computational analysis as carefully as the experiments, with the right controls and methods, and then write a manuscript that presents this in a concise but readable way.

Reviewer #5 (Remarks to the Author): Expert in single-cell RNA-seq and computational genomics

The manuscript is comprehensive, well written, and provides a useful data resource in a cancer type that is under-represented in recent cellular genomics work.

I strongly agree with the suggestion to include spatial data. This would be of significantly benefit to this work. Even on a small number of patient samples, it would provide an important way of validating conclusions regarding cell-cell interactions (from cellphonedb) identifying new biology, and help understand the potential to target the proposed gene/pathways derived from epithelial programs that possibly affect patient survival.

One other point, is inter-individual variation. In my opinion, the sample size (60 patients) is a significant strength of this study. However, data is analysed and collectively integrated. Examining the variation in cell proportions, expression of key genes, and co-regulation of pathways between individuals would provide useful information on why patients vary in their clinical features and disease progression. If data exists, correlating such variation with clinical features would be really interesting. Intra- and Inter tumoral heterogeneity are hypothesized to be one of the major drivers of variation in cancer survival and response to treatment.

Responses to the Reviewers' Comments

Reviewer #1

Authors have addressed all my concerns.

Response: We thank the Reviewer.

Reviewer #3

The authors have adequately performed targeted deep sequencing of ESCC samples to address my initial concerns.

Response: We thank the Reviewer.

Reviewer #4

Zhang et al present the data gathered from single-cell RNA sequencing of multiple esophageal squamous cell carcinoma samples, and a few normal controls.

While I am impressed by the extent of the created dataset, I am sad to say that I find the paper virtually unreadable. This is not primarily because of the abundant grammar and language issues, although they do not help. Rather, the presented analyses lack any kind of coherent hypothesis or structure. It forces me to conclude that this manuscript mainly constitutes a descriptive report of everything the authors could think of doing with the data they gathered. If this is the case, they should consider submitting this manuscript to something like "Scientific Data" or a similar journal.

If, however, their aim is to present a biological finding, then this needs to be rewritten from the ground up, and possibly broken into multiple papers. This is also because each individual sub-analysis is done superficially. I frankly refuse to go through each of the 40 pages of this paper and list every single bioinformatics issue, but just to name a few: they fail to even describe how unsupervised clustering was performed. There is no discussion on an evaluation of robustness of clustering. The described approach for evaluating inter- and intra-tumour heterogeneity seems highly dubious - there is an obvious confounding effect when using a 2d-projection like PCA to evaluate within- and between-patient distance at the same time. Their classification of "modules" of expression seems dubious to me. The classification of epithelial cells into "Group 1" and "Group 2" seems arbitrary. There is no discussion on the effect of choosing other arbitrary thresholds of separation.

This is by no means an exhaustive list of my grievances with this paper. More importantly, though, it is futile to try and "fix" all these issues until it's clear what the point is that the authors are really trying to make. They should focus on that, do the computational analysis as carefully as the experiments, with the right controls and methods, and then write a manuscript that presents this in a concise but readable way.

Response: Thanks for the comments and criticisms. Indeed, our manuscript has presented a large set of scRNA-seq data on ESCC; however, it is not to simply collect cell types and marker genes. The aim of our study was to characterize the heterogeneity among individual ESCC and to discover the important interactions among cancer cells and various other cells in the tumor microenvironment (TME) based on a large sample size. The manuscript has been logically organized in accordance with this aim. Firstly, we have analyzed the expression programs of esophageal epithelial cells (Figure 2 and 3) to show the highly inter-tumor heterogeneity. Secondly, we have carefully characterized

immune cells and non-immune stroma cells (Figure 4 and 5) to show the complicate ESCC TME and the interplays among these cells or with cancer cells, which has seldom been reported in ESCC and most other cancer types. These data are very important for the further investigations. Thirdly, we have integrated all the data to create the important associations between cancer cells and other types of cells in TME and identified the importance of the mucosal immunity-like (Mucosal) program in ESCC. Finally, we have discovered and verified the clinical relevance of 3 genes (i.e., *AGR2*, *CXCL17* and *MUC20*) in mucosal immunity program.

Thus, we believe that these results have made an important and logical story and should keep as a whole in one paper. We hope the Editor will make the decision.

Specific Comments

Comment 1: They fail to even describe how unsupervised clustering was performed.

Response 1: Per the comment, we have added a paragraph to Method of revised manuscript to descript how unsupervised clustering was performed (page 24, Lines 578–593) and hope this is satisfactory.

Comment 2: There is no discussion on an evaluation of robustness of clustering.

Response 2: Thanks for the comment. To validate the robustness of clustering, we compared the clusters identified using different number of principle components (PCs = 10 and 30) and resolutions (e.g., Res = 0.3 and 0.6) for major cell types including Endothelial, Fibroblast, Myeloid and T cells. We found most clusters are highly consistent across multiple clustering parameters except some minor confusions between adjacent clusters (see Figure below). Thus, we believe our cell clusters in the manuscript are robust and represent true biologically informative cell populations.

Per the comment, we have added a paragraph to the Methods section to describe robustness of clustering (lines 578 to 593).

Comment 3: The described approach for evaluating inter- and intra-tumour heterogeneity seems highly dubious - there is an obvious confounding effect when using a 2d-projection like PCA to evaluate within- and between-patient distance at the same time.

Response 3: The heterogeneity scores are essentially the quantification of similarities or distances among cell groups. In the fields of scRNA-seq analysis, cell clustering (Louvain and SLM algorithm) and visualization (tSNE and UMAP) are mainly based on Euclidean distance in PCA space. Thus, use of PCA projection for quantifying heterogeneity scores is legitimate. However, the number of dimensions or principal components (PCs) adopted for calculation might be an issue. In our original manuscript, the evaluations of inter- and intra-tumor heterogeneity were based on the top 20 PCs rather than the 2 dimensions as illustrated in the diagram of Figure 2d. To identify the suitable number of PCs more precisely, we have demonstrated the correlation of intra-tumor and inter-tumor heterogeneity calculated using number of PCs ranging from 10 to 50 (see Figures below). The color and number in each cell in the following figure represent the correlation coefficient of heterogeneity scores calculated using the corresponding two sets of number of PCs.

Supplementary Fig. 2 The heterogeneity and expression programs of epithelial cells.

(c) Pearson correlation coefficients for intra- and inter-tumor heterogeneity level using distances calculated from different number of PCs.

As shown, the intra-tumor heterogeneity scores are highly robust to number of PCs and inter-tumor heterogeneity scores are robust and stable for PC number exceeding 15. We have found that using PCs of 30 gave satisfactory results showing high correlation coefficient with more numbers of PCs ($r \geq 0.98$). Accordingly, we adopted 30 PCs for further analyses. We have added these results to revised manuscript (Supplementary Fig. 2c).

Comment 4: There is no discussion on the effect of choosing other arbitrary thresholds of separation.

Response 4: We are sorry that we did not discussed the thresholds in our original manuscript. There are two thresholds in the classification of Group 1 and Group 2 samples: cluster threshold and sample threshold (see Figure below). We first calculated the ratio of cells contributed by sole/single sample using a cluster threshold of 0.75 in our original manuscript, which generated 24 Group 1 and 14 Group 2 clusters (upper panel in the Figure below). Subsequently, we calculated the ratio of cells from the 24 Group 1 clusters for each sample, using sample threshold of 0.6, which identified 21

Group 1 samples. Because 15 of the 21 Group 1 samples using 30 PCs were identical to those defined previously by using 20 PCs, the rest 6 Group 1 samples were considered newly defined.

Although Group 1 and Group 2 clusters and samples are obviously dichotomy as shown in above Figures, the classification of some intermediate clusters/samples could be altered by using different thresholds. To justify the thresholds, we have performed analysis by combination of both cluster and sample thresholds from 0.4 to 0.9 at the step size of 0.05. To demonstrate the robustness of our threshold (0.75 for cluster and 0.6 for sample), we have calculated the ratio of samples in overlap with the 21 Group 1 sample for each cluster and sample threshold set (see Figure below). The result shows that under very loose thresholds such as 0.4 for cluster and 0.4 for sample, 70% of the Group 1 samples match to the threshold of 0.75 for cluster and 0.6 for sample, indicating that the Group 1 samples we identified are robust to the choice of thresholds of separation. We have added these results in revised manuscript (page 5, Lines 114–123; page 27, Lines 655-660).

Comment 5: Their classification of "modules" of expression seems dubious to me.

Response 5: Gene modules defined in our original manuscript were a set of genes co-expressed and coherently co-varying within subsets of epithelial cells. The shared or recurrent gene modules across multiple samples were aggregated into multiple expression programs. These highly concordant modules in different tumors may reflect the common patterns of inter-tumoral expression

heterogeneity (Puram et al., Cell 2017;171:1611–24). These classification and definition have been used in many single cell tumor cell studies. For example, it assisted the identification of functional significance of sub-population of malignant cells (Puram et al., Cell 2017;171:1611–24, Gojo et al., Cancer Cell 2020;38:44–59) and assessing immunoreactivity and stemness of tumors (Izar et al., Nat Med 2020;26:1271–9). To derive these programs, we have performed clustering within all epithelial cells from each sample and the results were highly similar to those analyzed by using Non-negative Matrix Factorization of expression matrix. Finally, we have identified 8 common programs using hierarchical clustering, many of which such as cell cycling, stress, antigen processing and epithelial differentiation programs have been reported in the previous studies mentioned above (Puram et al., Cell 2017;171:1611–24, Gojo et al., Cancer Cell 2020;38:44–59 and Izar et al., Nat Med 2020;26:1271–9). Interestingly, we have found a novel expression programs named “mucosal immunity” that has been analyzed more detail in this study. These results reflect the fact that our classification of "modules" of expression is reliable.

Reviewer #5

Comment 1: The manuscript is comprehensive, well written, and provides a useful data resource in a cancer type that is under-represented in recent cellular genomics work. I strongly agree with the suggestion to include spatial data. This would be of significantly benefit to this work. Even on a small number of patient samples, it would provide an important way of validating conclusions regarding cell-cell interactions (from cellphonedb) identifying new biology and help understand the potential to target the proposed gene/pathways derived from epithelial programs that possibly affect patient survival.

Response 1: We thank and totally agree with the suggestion. However, in view of already having huge data in this manuscript, which has been criticized by some Reviewers, we would beg not to include the spatial transcriptomics data in the current paper. We would like to present the spatial data in next paper.

Comment 2: One other point is inter-individual variation. In my opinion, the sample size (60 patients) is a significant strength of this study. However, data is analyzed and collectively integrated. Examining the variation in cell proportions, expression of key genes, and co-regulation of pathways between individuals would provide useful information on why patients vary in their clinical features and disease progression. Is data exists, correlating such variation with clinical features would be really interesting.

Response 2: Thanks for the comment. We have performed many correlation analyses. Firstly, we have analyzed the correlations among subtypes of cells in tumor microenvironment (TME). For example, as shown in Figure 5l, we have categorized immune cells into benign or malignant groups and show their diverse correlations with stromal cells, suggesting the interactions among cell types in TME. Secondly, for each cell type, we have analyzed its association with clinical features of patients including age, sex, drinking status, smoking status, somatic mutations and disease stage. These results have been presented in Figures 4b, 5e, 6b, S2d, S3d, S5o, S6d, and S6f. Furthermore, we have shown that the epithelial expression programs are associated with some cell types in TME and somatic mutations (Figures 6a and 6b), suggesting an interaction between tumor and the TME. We have also examined the correlations between the expression levels of some key genes in epithelial expression programs and patients' survival time and identified 3 significantly associated genes (*AGR2*, *CXCL17* and *MUC20*) in mucosal immunity program (Figure 6c-f). We believe that these analyses have provided a comprehensive framework for exploring the inter-individual variations and the results are interesting and promising.

Comment 3: Intra- and Inter tumoral heterogeneity are hypothesized to be one of the major drivers of variation in cancer survival and response to treatment.

Response 3: Yes. We agree with the comment. In this study, we have found that the relative heterogeneity levels are associated with patients' age and alcohol drinking status, two well-known risk factors for the development of ESCC. We have also found that the differences in the expression levels of *AGR2*, *CXCL17* and *MUC20* genes in mucosal immunity program are associated with survival time in patients. These results are in line with the hypothesis that the Reviewer mentions in the comment. Unfortunately, because the subjects recruited in this study were all only received surgery but not chemotherapy or radiotherapy, no treatment responses could be included in the analysis.

REVIEWER COMMENTS

Reviewer #5 (Remarks to the Author):

The authors have addressed the comments that I made

Responses to the Reviewers' Comments

Reviewer #4

Comment 3: The described approach for evaluating inter- and intra-tumour heterogeneity seems highly dubious - there is an obvious confounding effect when using a 2d-projection like PCA to evaluate within- and between-patient distance at the same time.

Response: Thanks for the comment. Heterogeneity reflects the differences among groups of cells and analyzing the correlations of gene expression is the most basic method to evaluate heterogeneity, i.e., different or similar. In our original manuscript, we used the PCA-based heterogeneity evaluations. Here we adopted a more fundamental correlation-based heterogeneity analysis in parallel. Specifically, we performed the correlation analysis on gene expression matrix of all epithelial cells for each sample and calculated the complement of averaged correlation coefficients as the correlation-based intra-tumor heterogeneity. We also performed the cross-correlation analysis between transcriptome data of cells from individual sample and all other cells and used the complement of averaged correlation coefficients as the correlation-based inter-tumor heterogeneity.

We have subsequently compared the correlation-based and the PCA-based heterogeneity scores as plotted in the following figures. We found that inter- and intra-tumor heterogeneity levels evaluated by the two approaches are quite similar ($r = 0.71$, $P = 3.2e-9$ and $r = 0.64$, $P = 2.9e-7$). These results suggest that the PCA-based analysis is able to evaluate within- and between-patient distances without obvious confounding effects.

Comparison between PCA-based and Correlation-based Heterogeneity Scores

However, differences exist for the two heterogeneity quantification approaches. The correlation-based method uses raw expression levels, while PCA-based method projects the original expression profiles of all malignant cells to the eigenvector space to derive principal components (PCs), which captures major information and reduce noise (Ma et al., Cancer cell 2019; 36:418-430). For both approaches, we have investigated the relationship of inter- and intra-tumor heterogeneity levels between Group 1 and Group 2 samples and found that Group 1 samples, as revealed in our manuscript, have relatively higher inter-tumor heterogeneity levels than Group 2 cluster at the same intra-tumor heterogeneity levels. We have shown that PCA-based heterogeneity scores distinguish better the differences of relative heterogeneity between Group 1 and Group 2 samples (please refer the following figures). Based on these results, we have concluded that the PCA-based heterogeneity scores can reliably reflect and quantify both inter-

and intra-tumour heterogeneity levels. We have added the information in revised manuscript (page 19, lines 446–449; page 27, lines 657–667; Supplementary Figs. 2c, d).

Comment 4: The analysis of “thresholds of separation” might not be replicated in new data.

Response: Thank for this potential question. In principle, the analysis identifies the relative similarity and distance of samples in the transcriptome space. Thus, it depends on the component of dataset and would be expected that the results might be different from one data to another as the reviewer presumed. However, our analysis pipeline is robust to different sample numbers. Since the new data is currently unavailable, we have downsampled our dataset and checked whether the results are consistent. We have randomly subsampled 40% ($n = 20$), 60% ($n = 30$) and 80% ($n = 40$) from all 52 samples for 100 times and processed using the same analysis method and thresholds (Cluster as 0.75 and Sample as 0.60) to separated them into Group 1 and Group 2 samples. For each sample, we have calculated its probability of being identified as Group 1 samples among all random samplings (please refer the following figures). As a result, in each downsampling analysis, the Group 1 samples identified have the highest possibility to be reliably classified. We also found that, with more samples, the difference between Group 1 and Group 2 samples are more remarkable, and thus it is more advocated using more samples to perform such analysis. These results indicated that, our analysis pipeline and thresholds are robust to series of sample numbers and sample compositions. The Group 1 samples we originally identified are still the most significant ones in different sampling analysis. Based on these analyses, we believe that in the case of new data, our analysis may also identify the most prominent Group 1 samples among them. We have added these results in revised manuscript (page 19, lines 446–449 and page 28, lines 673–676).

Comment 5: Their classification of "modules" of expression seems dubious to me.

Response 5: As addressed in our previous response to the same comment in the last round of review, we have validated our analytical pipelines and the results by using these pipelines are in accordant to many recent studies. To adequately justify the classification of expression modules, we have again constructed the gene co-expression network of all epithelial cells (see Figures below). The modules show high correlations co-expressed within all samples, indicating that they are likely to be biologically co-regulated. We have also performed gene ontology analysis and the results show that each gene module has different functions, but the similar functional pathways are assembled within a specific module (see Figure below). Based on these findings, we believe that our classification of the expression modules is reliable and reasonable. We have added these results in revised manuscript (page 7, lines 144–146; Supplementary Fig. 2g).